# UNIFIED DISCRETE DIFFUSION FOR SIMULTANEOUS VISION-LANGUAGE GENERATION

**Minghui Hu, Chuanxia Zheng[†], Tat-Jen Cham & P. N. Suganthan[‡]**
Nanyang Technological University, [†]University of Oxford, [‡]Qatar University
{e200008, ASTJCham}@ntu.edu.sg,
cxzheng@robots.ox.ac.uk, p.n.suganthan@qu.edu.qa

**Zuopeng Yang[§], Heliang Zheng, Chaoyue Wang & Dacheng Tao**
JD Explore Academy, [§]Shanghai Jiao Tong University
yzpeng@sjtu.edu.cn, zhengheliang@jd.com,
chaoyue.wang@outlook.com, dacheng.tao@gmail.com

## ABSTRACT

The recently developed discrete diffusion models perform extraordinarily well in the text-to-image task, showing significant promise for handling the multi-modality signals. In this work, we harness these traits and present a unified multimodal generation model that can conduct both the "modality translation" and "multi-modality generation" tasks using a single model, performing text-based, image-based, and even vision-language simultaneous generation. Specifically, we unify the discrete diffusion process for multimodal signals by proposing a unified transition matrix. Moreover, we design a mutual attention module with fused embedding layer and a unified objective function to emphasise the inter-modal linkages, which are vital for multi-modality generation. Extensive experiments indicate that our proposed method can perform comparably to the state-of-the-art solutions in various generation tasks.

## 1 INTRODUCTION

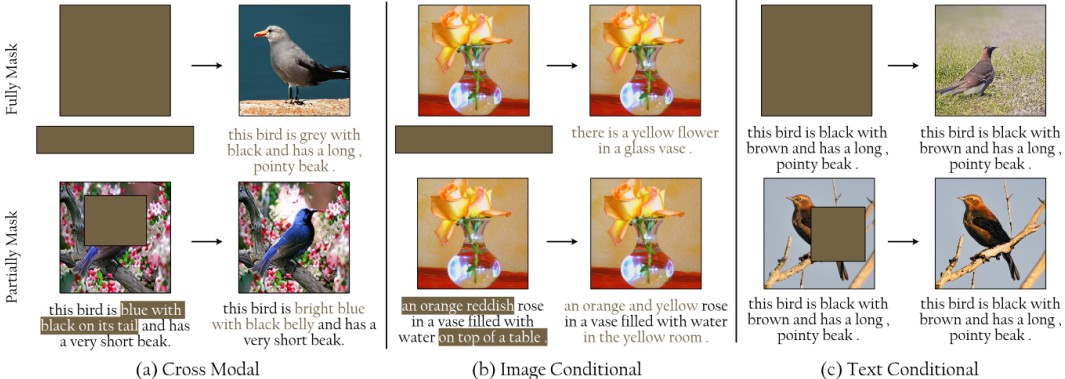

Figure 1: Examples of various tasks supported by UniD3. The dark brown portions of the image and description represent the [MASK].

Diffusion models (Ho et al., 2020; Song et al., 2021b) have garnered significant interest on various high quality conditional image generation tasks, such as image super-resolution (Rombach et al., 2022), image inpainting (Lugmayr et al., 2022), image editing (Avrahami et al., 2022), image translation (Saharia et al., 2022a), among others. Concurrently, the Vector Quantized (VQ) models have also achieved rapid advances in image generations, especially on cross-modal tasks, examples in-

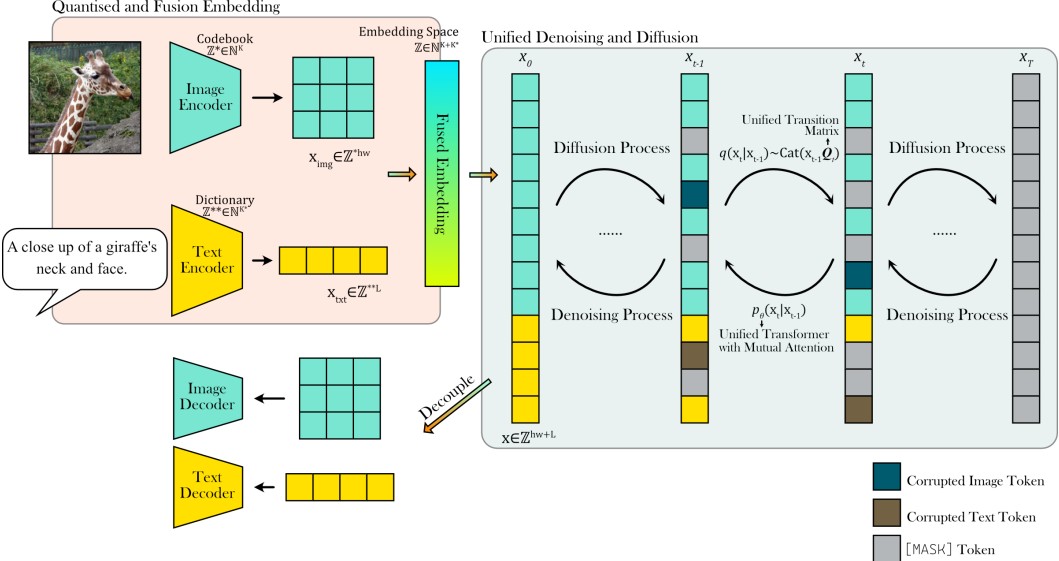

Figure 2: The pipeline of UniD3. With an offline model (red background part), the given inputs are represented by discrete token sequence in separate domain. The fusion embedding concatenate the tokens in different modal and embed them to the same space. The unified diffusion (in blue background) will construct the joint distribution of all modalities based on the fused embedding with a fixed unified Markov transition matrix.

clude text-to-image (Kim et al., 2022), sketch-to-image (Esser et al., 2021b), image-to-video (Wu et al., 2021).

Despite their success, all of these generation tasks are designed for only a single special modality, *i.e.* either *modality translation* or *modality generation*, with the help of a powerful diffusion model. In particular, the former *modality translation* translates the given conditional signals into the traget domain, while the latter *modality generation* only works for the unconditional image generation, *e.g.* image generation on CelebA Karras et al. (2017) or LSUN Yu et al. (2015) datasets. However, none of them consider to learn a join distribtuion for the mixture modality.

Here, we take the use of discrete diffusion model into a new realm - *multi-modality generation* using a *unified vision-language model*. In contrast to the *modality translation* tasks outlined above, our multi-modality generative model does not require any conditional signals given in prior and is capable of simultaneously generating content pairs with the associated multi-modalities. *UniD3*, our new **Uni**fied **D**iscrete **D**enoising **D**iffusion model, allows us to construct a joint vision-language probability distribution by mixing discrete image tokens with text tokens, leading to a capability of simultaneously generating cross-domain results. This is achieved by the two-stages framework, illustrated in Fig. 2, (1) An offline model to generate a compact yet expressive discrete representation for both images and texts (the pink part in Fig. 2). (2) A novel unified discrete diffusion model to estimate the joint distribution of such latent visual and language codes (the cyan part in Fig. 2). Once trained, *UniD3* can not only inherit the ability to manipulate the provided text or image, but is also able to unify the text and image generation, *e.g.*, unconditional vision-language pairings generation, cross modal manipulation, text guided image completion, and image conditional text caption (Fig. 1 depicts the tasks that our model is capable of handling).

Based on the empirical exploration, our model can achieve comparable image quality to the text condition in the unconditional case. In terms of image caption, our model is also comparable to some states of the art.

In summary, our key contributions include the following:

- We design a specific Markov transition matrix for our unified discrete denoising diffusion model, which lead to a sophisticated control of diffusion process, to estimate the joint

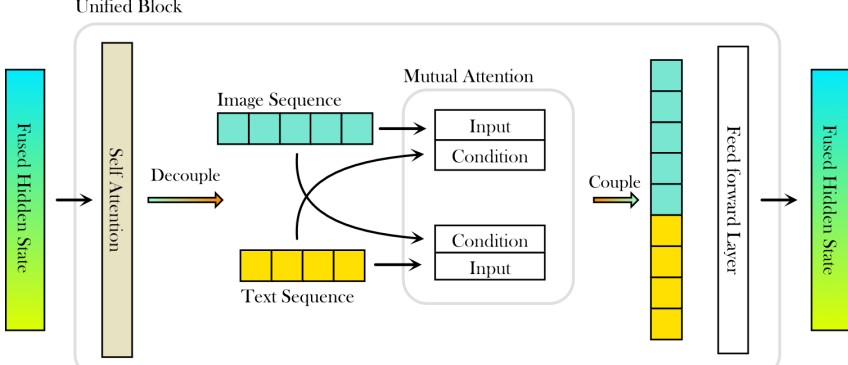

Figure 3: Illustration of transformer blocks with mutual attention. A unified transformer is composed of several blocks stacked on top of one another.

distribution of language and image. The purposive design of transfer matrices based on task objectives and data properties is also pioneering for discrete diffusion models.

- We additionally propose mutual mechanism with fuse embedding to fulfil the objective of multi-modal integration. And we alter a unified objective function to offer the optimization with more concise restrictions.

- To the best of our knowledge, the UniD3 is the first work to solve both *multi-modality generation* and *modality translation* problems that is capable of handling the simultaneous unconditional vision-language generation and bi-directional vision-language synthesis tasks with a single model.

## 2 PRELIMINARIES

**Vector Quantised Model** The Vector-Quantised Variational Auto Encoder (VQ-VAE) (Van Den Oord et al., 2017) learns to embed the high-dimensional data, e.g., image or audio, into a discrete representation. In particular, given a high dimensional data $\boldsymbol{x} \in \mathbb{R}^{C \times H \times W}$, the encoder $E$ first converts it to the spatial latent features $\boldsymbol{z} = \{z_{i,j}\} \in \mathbb{R}^{d \times h \times w}$, and then transfer this continuous features into discrete space by looking up the closest features in the codebook $\mathcal{Z} = \{z_k\} \in \mathbb{R}^{K \times d}$ to obtain the tokens $\boldsymbol{z}_q$:

$$\boldsymbol{z}_q = \text{Quantise}(\boldsymbol{z}) = \text{Quantise}(E(\boldsymbol{x})) := \arg\min_k \|z_{i,j} - z_k\|, \tag{1}$$

where the dimensions $h, w$ of latent feature $\boldsymbol{z}$ are substantially smaller than the original dimensions $H, W$. The reconstructions can be obtained through a decoder $G$:

$$\hat{\boldsymbol{x}} = G(\boldsymbol{z}_q) = G(\text{Quantise}(E(\boldsymbol{x}))). \tag{2}$$

Recently, there have been significant advances in learning more compact representation, yet higher-quality reconstruction, such as introducing new losses (Esser et al., 2021a), applying powerful backbone (Yu et al., 2021), using multiple channels representation (Lee et al., 2022) and spatially normalization (Zheng et al., 2022). However, in this paper, we focus on applying the technology in our novel UniD3, rather than exploring a new codebook learning approach.

**Discrete Diffusion** The discrete diffusion model was originally mentioned in (Sohl-Dickstein et al., 2015), with transitions converging to a binomial distribution. Subsequent work extended this to multinomial diffusion (Hoogeboom et al., 2021; Song et al., 2021a), while Austin et al. (2021) provided more options for transition matrices. Recent works integrated discrete diffusion models with VQ-VAE, allowing for high-quality image synthesis (Gu et al., 2022; Hu et al., 2022).

Here we briefly describe the multinomial diffusion with the absorbing state, as employed in VQ-Diffusion (Gu et al., 2022). Besides $K$ tokens from a discrete VAE, an additional [MASK] token is introduced. The forward process is defined as:

$$q(\boldsymbol{x}_t|\boldsymbol{x}_{t-1}) = \text{Cat}(\boldsymbol{x}_t; \boldsymbol{p} = \boldsymbol{Q}_t\boldsymbol{x}_{t-1}) = \boldsymbol{x}_t^T \boldsymbol{Q}_t \boldsymbol{x}_{t-1}, \tag{3}$$

where $\boldsymbol{x}$ is a one-hot vector identifying the token index. Here $[\boldsymbol{Q}_t]_{i,j} = q(x_t = i | x_{t-1} = j) \in \mathbb{R}^{(K+1)\times(K+1)}$ is the Markov transition matrix from $t-1$ to $t$, which can be expressed as:

$$
\boldsymbol{Q}_{[t-1 \to t]} = \begin{bmatrix} \alpha_t + \beta_t & \beta_t & \beta_t & \cdots & 0 \\ \beta_t & \alpha_t + \beta_t & \beta_t & \cdots & 0 \\ \beta_t & \beta_t & \alpha_t + \beta_t & \cdots & 0 \\ \vdots & \vdots & \vdots & \ddots & \vdots \\ \gamma_t & \gamma_t & \gamma_t & \cdots & 1 \end{bmatrix},
\tag{4}
$$

where $\alpha_t \in [0, 1]$ is the probability of retaining the token, and has a probability of $\gamma_t$ to be replaced by the [MASK] token, leaving a chance of $\beta_t = (1 - \alpha_t - \gamma_t)/K$ to be diffused.

The posterior of the diffusion process can be formulated as:

$$
q(\boldsymbol{x}_{t-1} | \boldsymbol{x}_t, \boldsymbol{x}_0) = \frac{q(\boldsymbol{x}_t | \boldsymbol{x}_{t-1}, \boldsymbol{x}_0) q(\boldsymbol{x}_{t-1} | \boldsymbol{x}_0)}{q(\boldsymbol{x}_t | \boldsymbol{x}_0)} = \frac{(\boldsymbol{x}_t^T \boldsymbol{Q}_t \boldsymbol{x}_{t-1})(\boldsymbol{x}_{t-1}^T \overline{\boldsymbol{Q}}_{t-1} \boldsymbol{x}_0)}{\boldsymbol{x}_t^T \overline{\boldsymbol{Q}}_t \boldsymbol{x}_0},
\tag{5}
$$

given $\overline{\boldsymbol{Q}}_t = \boldsymbol{Q}_t \cdots \boldsymbol{Q}_1$, which can be calculated in closed form, and

$$
q(\boldsymbol{x}_t | \boldsymbol{x}_0) = \mathrm{Cat}(\boldsymbol{x}_t; \boldsymbol{p} = \overline{\boldsymbol{Q}}_t \boldsymbol{x}_0) = \boldsymbol{x}_t^T \overline{\boldsymbol{Q}}_t \boldsymbol{x}_0.
\tag{6}
$$

In the reverse process, instead of explicitly predicting the posterior using a denoising neural network, the $\boldsymbol{x}_0$-parameterisation might increase the stability and permit fast inference (skipping $\Delta t$ steps per iteration). The reverse transition with reparameterisation is given as:

$$
p_\theta(\boldsymbol{x}_{t-1} | \boldsymbol{x}_t) \propto \sum_{\tilde{\boldsymbol{x}}_0} q(\boldsymbol{x}_{t-1} | \boldsymbol{x}_t, \tilde{\boldsymbol{x}}_0) \tilde{p}_\theta(\tilde{\boldsymbol{x}}_0 | \boldsymbol{x}_t),
\tag{7}
$$

in which the neural network predicts the logits of the target data $q(\boldsymbol{x}_0)$.

## 3 METHOD

Our goal is to adapt the discrete diffusion model to learn the joint distribution of linguistic and visual features concurrently. First, we propose a transition matrix that allows the diffusion model to capture the implicit association between text and images. Second, we present a mutual attention transformer architecture with fuse embedding layer as the denoising function and a unified objective function, which fits our unified diffusion process objective and permits for more precise predictions.

Our overall pipeline is illustrated in Fig. 2. Specifically, our solution begins by compressing the figures and texts into discrete token sequences using dVAE and BPE, respectively. A robust diffusion model with unified transition matrix is then constructed to fit the joint distribution across different modalities, which is further empowered by a transformer with a mutual attention mechanism.

### 3.1 UNIFIED DIFFUSION PROCESS

**Unified Transition Matrix** Discrete diffusion models with transition matrices can capture global links. The presence of a transition matrix determines the nature of the discrete diffusion model, which also provides us with more choices for token evolution. Thus we may wonder if it is feasible to design transition matrices that capture the global connections between various modalities.

The Markov transition matrix of the discrete diffusion model should satisfy the following requirements: 1. each column in $\boldsymbol{Q}_t$ should sum to one to conserve probability mass; 2. each column in the cumulative-product $\overline{\boldsymbol{Q}}_t$ should converge to either a known stationary distribution or a learnt prior when $t$ becomes large.

On the basis of these criteria, we construct a unified transition matrix $[\boldsymbol{Q}_t]_{i,j} = q(x_t = i | x_{t-1} = j)$ capable of encapsulating discrete representations among various modalities. The following matrix $\boldsymbol{Q}_t \in \mathbb{R}^{(K+K^*+1)\times(K+K^*+1)}$ illustrates a unified transition process with only text and image

modalities:

$$
\boldsymbol{Q}_t =
\left[
\begin{array}{ccccc|ccccc|c}
\alpha_t + \beta_t & \beta_t & \beta_t & \cdots & \beta_t & 0 & 0 & 0 & \cdots & 0 & 0 \\
\beta_t & \alpha_t + \beta_t & \beta_t & \cdots & \beta_t & 0 & 0 & 0 & \cdots & 0 & 0 \\
\vdots & \vdots & \vdots & \ddots & \vdots & \vdots & \vdots & \vdots & \ddots & \vdots & \vdots \\
\beta_t & \beta_t & \beta_t & \cdots & \alpha_t + \beta_t & 0 & 0 & 0 & \cdots & 0 & 0 \\
\hline
0 & 0 & 0 & \cdots & 0 & \alpha_t + \beta_t^* & \beta_t^* & \beta_t^* & \cdots & \beta_t^* & 0 \\
0 & 0 & 0 & \cdots & 0 & \beta_t^* & \alpha_t + \beta_t^* & \beta_t^* & \cdots & \beta_t^* & 0 \\
\vdots & \vdots & \vdots & \ddots & \vdots & \vdots & \vdots & \vdots & \ddots & \vdots & \vdots \\
0 & 0 & 0 & \cdots & 0 & \beta_t^* & \beta_t^* & \beta_t^* & \cdots & \alpha_t + \beta_t^* & 0 \\
\hline
\gamma_t & \gamma_t & \gamma_t & \cdots & \gamma_t & \gamma_t & \gamma_t & \gamma_t & \cdots & \gamma_t & 1
\end{array}
\right],
\tag{8}
$$

where $\alpha_t \in [0, 1]$ is the probability to keep this token, $\beta_t$ and $\beta_t^*$ are the probabilities of a token to be replaced by any other accessible tokens in different modality, and $\gamma_t$ is the absorbing probability.

The dimension of the matrix $\boldsymbol{Q}_t$ is $(K + K^* + 1) \times (K + K^* + 1)$, where $K$ and $K^*$ are the number of states in different modals respectively, e.g., $K$ is the size from the codebook in discrete VAE and $K^*$ is the dictionary size of BPE.

The matrix comprises five sections:

- The final row and column form a section associated with the transition of the absorbing state. Intuitively, if the token is [MASK] at $t - 1$ step, then the token must be [MASK] at time $t$. Conversely, any other modal token has the equal possibility $\gamma_t$ of being diffused to [MASK].

- The remainder of the matrix is composed of four quadrants, the first and third of which are fully zeros. Specifically, these two sub-matrices prevent tokens transitioning from one modality to another. The form of the second and fourth quadrants closely resembles that for multinomial diffusion. Here, in addition to some probability of being converted into a [MASK] token, each token also has some chance of transiting to a different state within the same modality, or remaining unaltered.

- The dimensions of these four quadrants are not identical, which are $K^* \times K$, $K \times K$, $K \times K^*$ and $K^* \times K^*$, respectively.

It is worth noting that $\alpha_t$ and $\gamma_t$ are the same in all modalities, whereas $\beta_t$ varies based on the number of states in different modalities. Mathematically, $\beta_t = (1 - \alpha_t - \gamma_t)/K$ and $\beta_t^* = (1 - \alpha_t - \gamma_t)/K^*$.

The sum of each column in this transition matrix is one to preserve probability mass, and also all the mass of the stationary distribution falls on the [MASK] token, which satisfies the prerequisite for a discrete diffusion model transition matrix.

The computation of $\overline{\boldsymbol{Q}}_t \boldsymbol{x}_0$, needed for $q(\boldsymbol{x}_t | \boldsymbol{x}_0)$ in Eq. 6, can be efficiently obtained in closed form:

$$
\overline{\boldsymbol{Q}}_t \boldsymbol{x}_0 = \overline{\alpha}_t \boldsymbol{x}_0 + \left[ \overline{\gamma}_t - \mathbb{1}(\boldsymbol{x}_0)\overline{\beta}_t - \mathbb{1}^*(\boldsymbol{x}_0)\overline{\beta}_t^* \right] \boldsymbol{x}_{[\text{M}]} + \mathbb{1}(\boldsymbol{x}_0)\overline{\beta}_t + \mathbb{1}^*(\boldsymbol{x}_0)\overline{\beta}_t^*,
$$

where $\mathbb{1}(\boldsymbol{x}_0) = \begin{cases} 1 & \text{if } \arg\max \boldsymbol{x}_0 \in [0, K), \\ 0 & \text{otherwise.} \end{cases}$, $\mathbb{1}^*(\boldsymbol{x}_0) = \begin{cases} 1 & \text{if } \arg\max \boldsymbol{x}_0 \in [K, K + K^*), \\ 0 & \text{otherwise.} \end{cases}$,

$\boldsymbol{x}_{[\text{M}]} = \boldsymbol{x} \leftarrow \arg\max \boldsymbol{x} = K + K^*$ and $\overline{\alpha}_t, \overline{\beta}_t, \overline{\beta^*}_t, \overline{\gamma}_t$ are the corresponding cumulative product.

$$\tag{9}$$

The detailed proof is provided in Appendix E.

**Unified Objective**    For conditional generation tasks such as text-to-image generation, the goal is to maximize $p(\boldsymbol{x}|\boldsymbol{y})$ by finding $\boldsymbol{x}$, where $\boldsymbol{y}$ is the given text condition. In our task, it can be approximated that the model try to maximize the joint distribution $p(\boldsymbol{x}, \boldsymbol{y})$ simultaneously. In practice, we minimize the Kullback-Leibler divergence between $q(\boldsymbol{x}_{t-1}|\boldsymbol{x}_t, \boldsymbol{x}_0)$ and $p_\theta(\boldsymbol{x}_{t-1}|\boldsymbol{x}_t)$ in both image and text directions, as shown in the following:

$$
\mathcal{L}_0 = -\mathbb{E}_{q(\boldsymbol{x}_1|\boldsymbol{x}_0)}[\log p_\theta(\boldsymbol{x}_0^{img}|\boldsymbol{x}_1, \boldsymbol{x}_0^{txt}) + \log p_\theta(\boldsymbol{x}_0^{txt}|\boldsymbol{x}_1, \boldsymbol{x}_0^{img})],
\tag{10}
$$

$$\mathcal{L}_{t-1} = \mathbb{E}_{q(\boldsymbol{x}_t|\boldsymbol{x}_0)} \left[ D_{\mathrm{KL}} \left( q(\boldsymbol{x}_{t-1}|\boldsymbol{x}_t, \boldsymbol{x}_0) \| \left[ p_\theta(\boldsymbol{x}_{t-1}^{img}|\boldsymbol{x}_t); p_\theta(\boldsymbol{x}_{t-1}^{txt}|\boldsymbol{x}_t) \right] \right) \right], \qquad (11)$$

where $p_\theta(\boldsymbol{x}_{t-1})$ is the integration of the logits $p_\theta(\boldsymbol{x}_{t-1}^{img})$ and $p_\theta(\boldsymbol{x}_{t-1}^{txt})$ from separate modal, and they can obtained with $\boldsymbol{x}_0$-parameterisation as following:

$$p_\theta(\boldsymbol{x}_{t-1}^{img}|\boldsymbol{x}_t) \propto \sum_{\tilde{\boldsymbol{x}}_0^{img}} q(\boldsymbol{x}_{t-1}^{img}|\boldsymbol{x}_t, \tilde{\boldsymbol{x}}_0^{img}, \boldsymbol{x}_0^{txt}) \tilde{p}_\theta(\tilde{\boldsymbol{x}}_0^{img}|\boldsymbol{x}_t)], \qquad (12)$$

$$p_\theta(\boldsymbol{x}_{t-1}^{txt}|\boldsymbol{x}_t) \propto \sum_{\tilde{\boldsymbol{x}}_0^{txt}} q(\boldsymbol{x}_{t-1}^{txt}|\boldsymbol{x}_t, \tilde{\boldsymbol{x}}_0^{txt}, \boldsymbol{x}_0^{img}) \tilde{p}_\theta(\tilde{\boldsymbol{x}}_0^{txt}|\boldsymbol{x}_t)]. \qquad (13)$$

Due to the $x_0$-parameterisation, our model is also capable of achieving fast sampling by increase the step size. And the last term of the variational lower bound loss $\mathcal{L}_T$ is a constant and can be ignored during the training:

$$\mathcal{L}_T = D_{\mathrm{KL}} \left( q(\boldsymbol{x}_T|\boldsymbol{x}_0) \| p(\boldsymbol{x}_T) \right), \qquad (14)$$

as the prior $p(\boldsymbol{x}_T)$ is fixed with the unified transition matrix:

$$p(\boldsymbol{x}_T) = \left[ \overline{\beta}_T, \overline{\beta}_T, \ldots, \overline{\beta}_T^*, \overline{\beta}_T^*, \ldots, \overline{\gamma}_T \right]. \qquad (15)$$

The full expression of the loss function can be found in Appendix C.

## 3.2 Denoising Function for Multimodal

**Mutual Attention** As indicated by Eqs. 12 & 13, the neural network is responsible for prediction of the distribution $\tilde{p}_\theta(\tilde{\boldsymbol{x}}_0|\boldsymbol{x}_t)$. However, the input to the neural network covers all modalities, and a simple self-attention mechanism can scarcely highlight the inter-modal linkages. In addition, the distributions predicted by the neural network need to be decoupled according to the various modalities throughout the reparameterisation. In other words, the network predictions should be expressed in terms of different modalities, e.g., $\tilde{p}_\theta(\tilde{\boldsymbol{x}}_0^{img}|\boldsymbol{x}_t)$ and $\tilde{p}_\theta(\tilde{\boldsymbol{x}}_0^{txt}|\boldsymbol{x}_t)$, with interconnections.

Therefore, we propose the mutual attention mechanism and construct the unified transformer as shown in Fig. 3. The unified transformer contains several transformer blocks, each of which consists of one self-attention, two parallel mutual attention operations and one feed-forward module. Each block receives a sequence of mixed-modal tokens as input that traverses a layer of self-attention to capture the inherent connection within the modalities. Afterwards, the hidden tokens are decoupled according to the location of the different modalities and fed to the corresponding mutual attention layers. Mutual attention is a modified version of the cross-attention layer, with the conditional inputs to cross-attention being maintained constant while the inputs to mutual attention are derived from hidden features. Next, the outputs from the various mutual attention are concatenated into one mixed-modal token sequence for transmission to the next block. At the end of the unified transformer, we layer-normalise the sequence of tokens from the blocks, which is then decoupled to different predictive distributions by fully-connected layers.

In our model, each modality may become a component of what needs to be generated. Therefore, we propose mutual attention to enable tokens of different modalities in a sequence to be conditional on each other, allowing the capture of the relationships between the various modalities. Mathematically, our mutual attention can be expressed as

$$\mathrm{MA}(T_i, T_j) = Attn(T_i, T_j; W) = \mathrm{softmax}(\frac{Q_i K_j^T}{\sqrt{2}}) V_j,$$

where $Q_i = W_Q T_i$, $K = W_K T_j$ & $V = W_V T_j$, and $T_i$ & $T_j$ are the tokens in a different modality, e.g. image and text. The Unified block contains a self-attention and several mutual attention in parallel. Given a hidden token vector $H' = [H'_i, H'_j]$, the whole pipeline of Unified block is

$$1. T' = SA(H') \quad 2. \ \text{Decouple:} \ T' \to T'_i, T'_j$$

$$3. \begin{cases} T_i = \mathrm{MA}(T'_i, T'_j) \\ T_j = \mathrm{MA}(T'_j, T'_i) \end{cases} \quad 4. \ \text{Couple:} \ T \leftarrow T_i, T_j$$

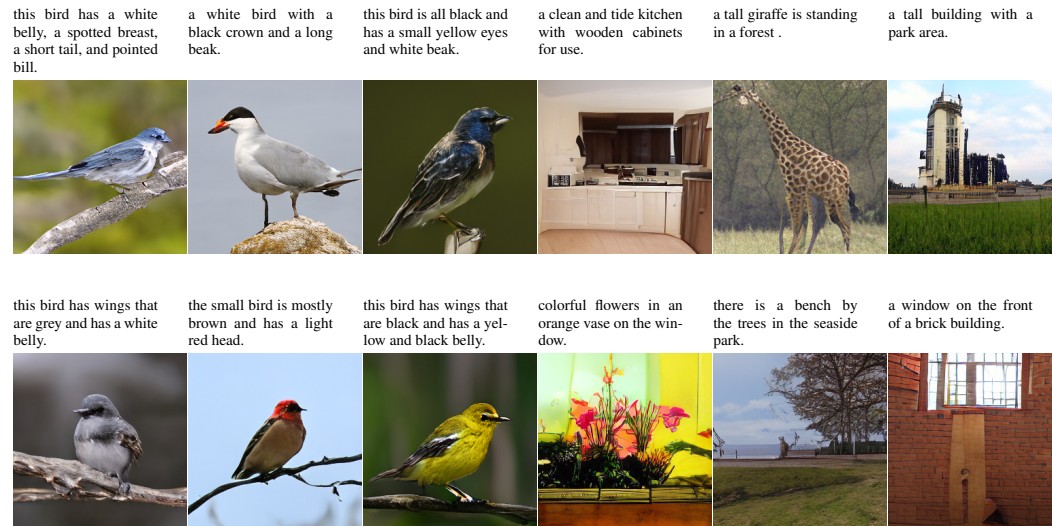

Figure 4: Generated vision-language Pairs from CUB-200 and MSCOCO. Both the image and caption are *generated simultaneously*. The quality of the created photos and text is comprehensible, and there is a correlation between the descriptions and the visuals.

**Fused Embedding**    In addition, discrete tokens of various modalities should be embedded in the same space. Considering that two modalities have $K$ and $K^*$ states respectively, we firstly create an embedding layer with size $K + K^* + 1$, with the additional state for the [MASK] token. We then create a learnable spatial position encoding for the image modality and a learnable sequence position encoding for the text modality. The final fused embedding is obtained by adding the embedded vectors to the positional encoding of their associated modalities.

## 4    EXPERIMENTS

The description of the datasets and model experimental details can be found in Appendix B.

### 4.1    QUANTITATIVE RESULTS

In the unconditional scenario, our model is able to sample from all modalities while maintaining the relationship between them. For conditional generation, we conducted experiments on text-based image generation and image-based caption generation, respectively.

**Metrics**    We evaluate the multimodal capacity of our model in three distinct areas: Image sampling quality, text sampling quality and the similarity between sampled image and text. Fréchet inception distance (FID) (Heusel et al., 2017) and Inception Score (IS) (Salimans et al., 2016) are used to evaluate the realism and variety of the produced images. The BLEU-4, METEOR and SPICE are used to assess the quality of a picture caption. And CLIP (Radford et al., 2021) is used to identify the similarity between visuals and text.

**Image Quality**    We quantitatively compare our unconditional and text-conditional results with several state-of-the-art text-to-image and bidirectional conditional generation methods, including GAN-based methods, diffusion-based methods and autoregressive methods. The comparison results are given in Table 1, in which *Pair* means the unconditional generated vision-language pairings and *T2I* represents the text-to-image generation.

The first section of the table solely covers modality translation methods. In the second part, these models are claimed to be endowed with cross modal capabilities. However, such methods are not able to achieve simultaneous vision-language pair generation and necessitate given signals. For a fair

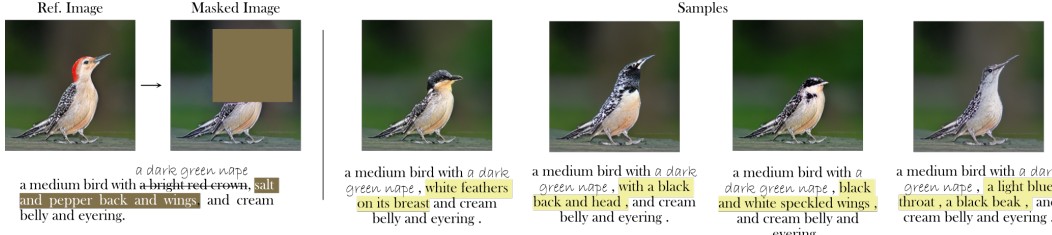

Figure 5: Presentation of the results of cross modal vision and language manipulation and infilling. The dark brown portions of the image and description represent the `[MASK]`, while the strike-through represent the caption manipulation. Image and text compliment each other simultaneously.

comparison, we exhibit the quantitative generative results based on various modalities individually. We provide more experimental results for text conditional generation in Appendix G.2

| | CUB | | MSCOCO | |
|---|---|---|---|---|
| | FID (↓) | IS (↑) | FID (↓) | IS (↑) |
| DM-GAN (Zhu et al., 2019) | 16.09 | 4.75 | 32.64 | 27.3 |
| DF-GAN (Tao et al., 2022) | 14.81 | 5.1 | 21.42 | - |
| VQ-Diffusion (Gu et al., 2022) | 11.94 | - | 19.75 | - |
| LDM w/o G (Rombach et al., 2022) | - | - | 23.31 | 20.23 |
| LAFITE(LF) (Zhou et al., 2022) | 27.53 | 4.32 | 18.04 | 27.2 |
| Unifying (Huang et al., 2021) | - | - | 29.9 | - |
| L-Verse (Kim et al., 2022) | - | - | 45.8 | - |
| OFA (Wang et al., 2022) | - | - | 10.5 | 31.1 |
| Ours(Pair) | 17.38 | 6.11 | 28.63 | 24.44 |
| Ours(T2I) | 16.19 | 6.02 | 25.11 | 21.03 |

Table 1: Evaluation of image synthesis on the CUB-200-2011 and MS-COCO dataset. The first section is the text-conditioned re-sults while the model in second part includes multimodal solution.

| | B-4 (↑) | M (↑) | S (↑) |
|---|---|---|---|
| BUTD (Anderson et al., 2018) | 36.3 | 27.7 | 21.4 |
| ORT (Herdade et al., 2019) | 38.6 | 28.7 | 22.6 |
| AoA (Huang et al., 2019) | 38.9 | 29.3 | - |
| X-LAN (Pan et al., 2020) | 39.7 | 29.5 | 23.4 |
| SimVLM (Wang et al., 2021) | 40.6 | 33.7 | 25.4 |
| OSCAR (Li et al., 2020) | 40.5 | 29.7 | 22.8 |
| Unifying (Huang et al., 2021) | 37.3 | 28.2 | 21.9 |
| L-verse (Kim et al., 2022) | 39.9 | 31.4 | 23.3 |
| OFA (Wang et al., 2022) | 41.0 | 30.9 | 24.2 |
| Ours(I2T) | 39.6 | 29.3 | 23.4 |

Table 2: Evaluation of image caption on the MSCOCO *Karpathy* split. The first section is the text-conditioned results while the model in second part includes multimodal solution.

**Text Quality**   In the case of image captions, we compared several other text description methods, the details are shown in Table 2, where *I2T* is the image-based text-generation task. Similarly, the first part of the table has pure image caption solutions and the bi-directional methods are demon-strated in the second part. Some results for image caption tasks are provided in Appendix G.1

**Similarity**   In the multimodal generation phase, all the other methods for comparison need the input of at least one specified modality; instead our method generates both visual and verbal features. The generated vision-language pairings are shown in Fig. 4 while more pair samples are provided in Appendix G.4. In this experiment, we use CLIP scores to evaluate the similarity between our generated visual and verbal modalities, while generated images with given texts were used for other methods. The comparison results are given in Table 3, where a higher correlation between the text/image samples of our model can be found.

### 4.2   CROSS MODAL MANIPULATION

To demonstrate the modification potential of our vision-language federation, we present the out-comes of picture alteration with description modification in Fig. 5 and more samples can be found in Appendix G.3. In this experiment, we obscured a portion of the image and modified the text description as well, the tokens in the masked region are merely designated as `[MASK]` tokens. We anticipate that the model will complete the image based on the amended text, in conjunction with the description based on the visible image region. Depending on the unmasked images, our model may supplement the caption that is derived from a portion of the image. Moreover, based on the amended description, the model may enhance the appearance of the masked component. These findings in-dicate that our model is capable of modifying text and pictures in both ways, in addition to jointly generating text and images.

## 4.3 ABLATION STUDY

In the ablation test, we utilized a rapid sampling scheme, with an interval between each step of 10. First, we evaluated the outcomes without mutual attention, where we replaced all mutual attention in the model structure with causal self-attention. In addition, we evaluated the performance without the unified transition matrix, we employed a matrix similar to Equation 4, with the out-of-bound outputs substituted by the minimal ordinal number of the corresponding modal.

| (↑) | CUB | COCO |
|---|---|---|
| Eval set | 0.286 | 0.306 |
| VQ-Diff (Gu et al., 2022) | - | 0.257 |
| iVQ-Diff (Tang et al., 2022) | - | 0.304 |
| Unifying (Huang et al., 2021) | - | 0.309 |
| OFA (Wang et al., 2022) | - | 0.344 |
| Ours(Pair) | 0.280 | 0.301 |
| Ours(T2I) | 0.293 | 0.306 |
| Ours(I2T) | 0.302 | 0.312 |

Table 3: The CLIP similarity between the generated captions and images.

| (↓) | FID |
|---|---|
| Ours | 17.38 |
| *w/* Fast Sampling $\Delta t = 10$ | 20.96 |
| *w/o* Mutual Attention | 23.19 |
| *w/o* Unified Transition Matrix | 32.63 |
| Ours (T2I) | 16.19 |
| *w/* Fast Sampling $\Delta t = 10$ | 18.33 |
| *w/o* Mutual Attention | 21.65 |
| *w/o* Unified Transition Matrix | 28.93 |

Table 4: Ablations studies. The experiments are conducted on the CUB-200.

## 5 RELATED WORK

**Denoising Diffusion Model**   Sohl-Dickstein et al. (2015) initially presented the diffusion model, which consists of a base Gaussian diffusion and a basic binomial diffusion. Ho et al. (2020) recently reignited interest in the extraordinary generating capacity of diffusion models. This is extended in subsequent research to enhance quality with the classifier guidance (Nichol & Dhariwal, 2021; Dhariwal & Nichol, 2021) and speed of sampling (Song et al., 2021a; Salimans & Ho, 2021). In addition to continuous diffusion models that have received the significant attention, there is also substantial research based on discrete diffusion models (Hoogeboom et al., 2021; Austin et al., 2021) for text generation or image segmentation. With the help of VQ-VAE, Esser et al. (2021a); Hu et al. (2022); Gu et al. (2022); Tang et al. (2022) discrete diffusion models have become capable of generating high quality images.

**Modality Translation**   Text-to-image generation is one of the key aspects of modality translation tasks. On simpler datasets, traditional GAN-based models (Xu et al., 2018; Zhu et al., 2019) may create high-quality pictures based on text hints, but face difficulties on more complex datasets. As a result of the emergence of transformers employing the attention mechanism (Vaswani et al., 2017), conventional autoregressive models (Van den Oord et al., 2016) have gained more powerful generating capacities as ART (Auto-regressive Transformers) (Ramesh et al., 2021; Esser et al., 2021b; Yu et al., 2022). Other types of modality translation models involve the addition of prospective modalities as criteria for improved generation outcomes (Gafni et al., 2022). There are also some works that focus on addressing bidirectional generation between different modalities (Kim et al., 2022; Huang et al., 2021; Wang et al., 2022); however, these solutions are task-specific, necessitating different input formats and specific models with distinct modalities.

## 6 CONCLUSION

This work presents UniD3, a novel unified framework for multi-modality generation tasks. We fisrtly introduced a unified transition matrix for the diffusion process which permits the states of fused tokens to be corrupted in a stable way. We demonstrated that designing transition matrices based on the objective and data characteristics is beneficial for discrete diffusion models. To capture the connection from the various modalities, we suggested a mutual attention with fuse embedding for noiseless state recovery. Our method is also generic and competent to perform modality translation tasks for the specified modality. The specific designs of transition matrix are intriguing for further exploring, as well as the performance of more modalities is also one of the future directions.

## 7 ETHICS

In this work, we propose UniD3, a new method for multimodal multi-task generation based on the discrete denoising diffusion model. All datasets evaluated in our experiments are open source, publicly accessible, and used with permission.

Similar to other generation algorithms, our method has both beneficial and negative societal effects, depending on its application and usage.

- Positively, UniD3 may explore the limits of deep learning generation through a unique pattern of graphic pair generation, while letting the typical user's imagination run wild and decreasing the entrance barrier for visual content production.
- On the other hand, UniD3 may be used to make modified photographs of great quality that are difficult to discern from the originals. Ways to fool humans and propagate false-hoods that are malicious. UniD3 can be abused by nefarious users, which may have serious repercussions.
- In our code distribution. We shall outline the permitted uses of our system and give the corresponding licences.

However, we observe that the present discriminators used to distinguish generated images are ineffective at identifying images generated by diffusion models. There is still the necessity for more exploration to discern between authentic and fake images.

## 8 REPRODUCIBILITY

In order to achieve reproducibility, we have made the following efforts:

1. The inference codes are released at https://github.com/mhh0318/UniD3.
2. Details on dataset and model architectures are provided in Appendix B, as well as more experimental settings are provided in Appendix D.
3. Some critical proofs are included in Appendix C & E.
4. Additional experimental results are provided in Appendix G.

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

## A  LIMITATIONS AND DISCUSSION

Our work focuses on multi-modality generation, and the task of every modal generation is taken into account. We consequently cannot employ adequate text or visual representations that are difficult to recover, which putting some pressure on the generative model in order to comprehend and represent the original data. This leads to a lack of performance of our model compared to the state-of-the-art methods in terms of conditional generation quality.

With the permission to manipulate the conditional space for a better representation, it can alleviate the burden on the generative model to represent the condition. Most of the current modality translation models use a highly abstract representation of a text as a conditional signal, for example, CLIP Radford et al. (2021) embedding are widely used in VQ-Diffuion Gu et al. (2022), Stable Diffusion / LDM Rakhimov et al. (2021). Some work has proved that a powerful text encoder (T5 Raffel et al. (2020)) and a highly abstract text representation could provide a better result Saharia et al. (2022b).

In a future challenge, we will consider obtaining a condensed modal representation as a discrete embedding, which is also simple to recover. This solution would reduce the strain on the generative model, enabling it to concentrate on producing superior generative results. We also found that the first-stage VQ model also partially limits the quality of our image generation. Finding a VQ model with a lower information loss rate is also an important way to improve the performance of the model.

Besides, our model demonstrates sufficient superiority in terms of text-image similarity without optimized by the CLIP loss, as shown in Table 3. In contrast, both improved VQ-Diffusion Tang et al. (2022) and OFA Wang et al. (2022) make explicitly or implicitly use of CLIP loss to optimise the model parameters.

## B  DATASETS AND EXPERIMENTAL DETAILS

### B.1  DATASETS

We demonstrate the feasibility of the proposed method on two commonly used datasets: CUB-200 (Wah et al., 2011) and MSCOCO (Lin et al., 2014). The CUB-200 dataset consists of 8,855 training images and 2,933 test images representing 200 species of birds. Each of these photos is accompanied by 10 textual descriptions. In MSCOCO, there are 591,753 images utilized for training and 25,014 for testing, with each image corresponding to 5 textual descriptions.

### B.2  MODEL DETAILS

We use VQ-GAN (Esser et al., 2021b) with gumbel softmax to compress the images into discrete token sequences. We directly use the pre-trained and public model which is trained on OpenImage (Krasin et al., 2017). The compression ratio of the model is $8 \times 8 \times 3 = 192$, and each image is compressed into a $32 \times 32$ token sequence. The codebook size is 2,887 after removing useless codes (Gu et al., 2022).

Furthermore, as the transition matrix of the diffusion model is sensitive to the number of dictionary size and the length of the sequence influences the performance of the denoising neural network, we further investigated the effect of various forms of text word lists. We select the dVAEs with different downsampling factors and dictionary size from (Esser et al., 2021b). We used alternative models of VQ-GAN release, including different downsampling factor with different codebook sizes, from $f8$ with 8,192 terms $\mathcal{Z} \in \mathbb{R}^{8192 \times 256}$ to $f16$ with 16,384 entries $\mathcal{Z} \in \mathbb{R}^{16384 \times 256}$.

For the text portion, we use a BPE tokenizer comparable to (Ramesh et al., 2021; Radford et al., 2021) and set the dictionary size to 8,192, compressing each text description into a sequence of length 128. The text and image encoders are fixed in the training phase.

### B.3  EXPERIMENTAL DETAILS

For the diffusion process, we set the number of diffusion steps to 500. And the noise planning is linear, where $\overline{\alpha_t}$ goes from 1 to 0 and $\overline{\gamma_t}$ goes from 0 to 1. The denoising network architecture is

| ($\downarrow$) | FID | Sample Steps |
|---|---|---|
| Ours | 17.38 | 500 |
| $w/$ $f$8-8192 Gumbel Image Encoder | 25.68 | 100 |
| $w/$ $f$16-974 Image Encoder | 21.02 | 100 |
| $w/$ $f$16-16384 Image Encoder | 24.61 | 100 |

Table 5: Ablations studies on Image Encoder. The experiments are conducted on the CUB-200.

as described in Sec.3.2, in which the transformer comprises 20 transformer blocks with 16 heads attention and 1024 feature dimension. The model contains 600M parameters. For the ablation model, we use 18 transformer blocks with 16 heads, and the dimension is 256. The model contains 119M parameters. The optimiser for the model is AdamW (Loshchilov & Hutter, 2018), and the learning rate is $9e^{-4}$ without warmup. We trained all models with a batch size of 16 across 8 Tesla A100.

It is worth noting in the comparison experiments that LAFITE (Zhou et al., 2022) is a language-free model. And LDM (Rombach et al., 2022) is trained on LAION-400M and conducts zero-shot text-to-image generation based on MSCOCO evaluation captions. The results of LDM is with a $f$8-KL based image encoder and fast sampling strategy, excluding the classifier guidance during inference.

And as the caption generation in our *Pair* scenario is unique. Common text generation metrics are difficult to implement due to a lack of the references. Here we used Perplexity (PPL) with GPT-2 to evaluate the text quality. The perplexities of generated CUB-200 and COCO captions are 123.32 and 188.74 whereas the corresponding values for the evaluation set are 108.66 and 175.36.

## C   LOSS FUNCTION

Similar to the continuous domain, the complete expression of the loss function in our model is:

$$
\begin{aligned}
\mathcal{L}_{vb} =& \mathbb{E}_{q(\boldsymbol{x}_0)} \left[ D_{\mathrm{KL}} \left( q(\boldsymbol{x}_T|\boldsymbol{x}_0) \| p(\boldsymbol{x}_T) \right) \right] \\
&+ \sum_{t=2}^{T} \mathbb{E}_{q(\boldsymbol{x}_t|\boldsymbol{x}_0)} \left[ D_{\mathrm{KL}} \left( q(\boldsymbol{x}_{t-1}|\boldsymbol{x}_t, \boldsymbol{x}_0) \| \left[ p_\theta(\boldsymbol{x}_{t-1}^{img}|\boldsymbol{x}_t); p_\theta(\boldsymbol{x}_{t-1}^{txt}|\boldsymbol{x}_t) \right] \right) \right] \\
&- \mathbb{E}_{q(\boldsymbol{x}_1|\boldsymbol{x}_0)} \left[ \log p_\theta(\boldsymbol{x}_0^{img}|\boldsymbol{x}_1, \boldsymbol{x}_0^{txt}) + \log p_\theta(\boldsymbol{x}_0^{txt}|\boldsymbol{x}_1, \boldsymbol{x}_0^{img}) \right].
\end{aligned}
\tag{16}
$$

## D   TRUNCATION SAMPLING

As stated in (Gu et al., 2022), the truncation sampling is crucial for the discrete diffusion-based approach, which prevents the network from sampling tokens with low probability. In our experiments, the truncation rate to 0.88 for *Pair* and *T2I* generation and 0.75 for *I2T* task, mathematically, we only keep top 88% or 75% tokens of $p_\theta(\tilde{\boldsymbol{x}}_0|\boldsymbol{x}_t)$ during inference for image and text tasks, respectively. A truncation rate that is too low will cause loss of image detail, but a rate that is too high will prohibit the image from creating a distinct geometry.

## E   PROOF OF THE CLOSED-FORM SOLUTION FOR $\overline{\mathbf{Q}}_t$

The solution for $\overline{\boldsymbol{Q}}_t$ inherits the closed-form attribute from (Gu et al., 2022). Mathematically, given the initial state $\boldsymbol{x}_0$ at $t = 0$, the probabilities for the next time-step $t = 1$ can be obtained:

$$
\overline{\boldsymbol{Q}}\boldsymbol{x}_0 = \begin{cases}
\overline{\alpha}_1 + \overline{\beta}_1, & \boldsymbol{x} = \boldsymbol{x}_0 \quad \textbf{s.t.}\, \operatorname{argmax} \boldsymbol{x}_0 \in [0, K), \\
\overline{\alpha}_1 + \overline{\beta}_1^*, & \boldsymbol{x} = \boldsymbol{x}_0 \quad \textbf{s.t.}\, \operatorname{argmax} \boldsymbol{x}_0 \in [K, K + K^*), \\
\overline{\beta}_1, & \boldsymbol{x} \neq \boldsymbol{x}_0 \quad \textbf{s.t.}\, \operatorname{argmax} \boldsymbol{x}_0 \in [0, K), \\
\overline{\beta}_1^*, & \boldsymbol{x} \neq \boldsymbol{x}_0 \quad \textbf{s.t.}\, \operatorname{argmax} \boldsymbol{x}_0 \in [K, K + K^*), \\
\overline{\gamma}, & \operatorname{argmax} \boldsymbol{x} = K + K^*.
\end{cases}
\tag{17}
$$

Suppose the closed-form expression of $\overline{\mathbf{Q}}_\tau$ holds at time-step $t = \tau$, then for the next step $t = \tau+1$:

$$\overline{Q}_{\tau+1}x_0 = Q_{\tau+1}\overline{Q}_\tau x_0. \tag{18}$$

The outputs under different conditions can be discussed:

1. When $x = x_0$ and $\operatorname{argmax} x_0 \in [0, K)$:

$$
\begin{aligned}
Q_{\tau+1}x_0 &= \overline{\beta}_\tau \beta_{\tau+1}(K-1) + (\alpha_{\tau+1} + \beta_{\tau+1})\left(\overline{\alpha}_\tau + \overline{\beta}_\tau\right) \\
&= \overline{\beta}_\tau\left(K\beta_{\tau+1} + \alpha_{t+1}\right) + \overline{\alpha}_t\left(\alpha_{t+1} + \beta_{t+1}\right) \\
&= \frac{1}{K}\left[\overline{\beta}_\tau(1 - \gamma_{\tau+1}) + \overline{\alpha}_\tau \beta_{\tau+1} - \overline{\beta}_{\tau+1}) * K\right] + \overline{\alpha}_{\tau+1} + \overline{\beta}_{\tau+1} \\
&= \frac{1}{K}\left[(1 - \overline{\alpha}_\tau - \overline{\gamma}_\tau)(1 - \gamma_{\tau+1}) + K\overline{\alpha}_\tau \beta_{\tau+1} - (1 - \overline{\alpha}_{\tau+1} - \overline{\gamma}_{\tau+1})\right] + \overline{\alpha}_{\tau+1} + \overline{\beta}_{\tau+1} \\
&= \frac{1}{K}\left[(1 - \overline{\gamma}_{\tau+1}) - \overline{\alpha}_\tau(1 - \gamma_{\tau+1} - K\beta_{\tau+1}) - (1 - \overline{\gamma}_{\tau+1}) + \overline{\alpha}_{\tau+1}\right] + \overline{\alpha}_{\tau+1} + \overline{\beta}_{\tau+1} \\
&= \overline{\alpha}_{\tau+1} + \overline{\beta}_{\tau+1};
\end{aligned}
\tag{19}
$$

2. When $x = x_0$ and $\operatorname{argmax} x_0 \in [K, K + K^*)$:

$$
\begin{aligned}
Q_{\tau+1}x_0 &= \overline{\beta^*}_\tau \beta^*_{\tau+1}(K^* - 1) + \left(\alpha_{\tau+1} + \beta^*_{\tau+1}\right)\left(\overline{\alpha}_\tau + \overline{\beta^*}_\tau\right) \\
&= \overline{\beta^*}_\tau\left(K^*\beta^*_{\tau+1} + \alpha_{t+1}\right) + \overline{\alpha}_t\left(\alpha_{t+1} + \beta^*_{t+1}\right) \\
&= \frac{1}{K^*}\left[\overline{\beta^*}_\tau(1 - \gamma_{\tau+1}) + \overline{\alpha}_\tau \beta^*_{\tau+1} - \overline{\beta^*}_{\tau+1}) * K^*\right] + \overline{\alpha}_{\tau+1} + \overline{\beta^*}_{\tau+1} \\
&= \frac{1}{K^*}\left[(1 - \overline{\alpha}_\tau - \overline{\gamma}_\tau)(1 - \gamma_{\tau+1}) + K^*\overline{\alpha}_\tau \beta^*_{\tau+1} - (1 - \overline{\alpha}_{\tau+1} - \overline{\gamma}_{\tau+1})\right] + \overline{\alpha}_{\tau+1} + \overline{\beta^*}_{\tau+1} \\
&= \frac{1}{K^*}\left[(1 - \overline{\gamma}_{\tau+1}) - \overline{\alpha}_\tau(1 - \gamma_{\tau+1} - K^*\beta^*_{\tau+1}) - (1 - \overline{\gamma}_{\tau+1}) + \overline{\alpha}_{\tau+1}\right] + \overline{\alpha}_{\tau+1} + \overline{\beta^*}_{\tau+1} \\
&= \overline{\alpha}_{\tau+1} + \overline{\beta^*}_{\tau+1};
\end{aligned}
\tag{20}
$$

3. When $x \neq x_0$ and $\operatorname{argmax} x_0 \in [0, K)$:

$$
\begin{aligned}
Q_{\tau+1}x_0 &= \overline{\beta}_\tau(\alpha_{\tau+1} + \beta_{\tau+1}) + \overline{\beta}_\tau \beta_{\tau+1}(K - 1) + \overline{\alpha}_\tau \beta_{\tau+1} \\
&= \overline{\beta}_\tau(\alpha_{\tau+1} + \beta_{\tau+1}) * K + \overline{\alpha}_\tau \beta_{\tau+1} \\
&= \frac{1 - \overline{\alpha}_\tau - \overline{\gamma}_\tau}{K} * (1 - \gamma_{\tau+1}) + \overline{\alpha}_\tau \beta_{\tau+1} \\
&= \frac{1}{K}(1 - \overline{\gamma}_{\tau+1}) + \overline{\alpha}_\tau(\beta_{\tau+1} - \frac{1 - \gamma_{\tau+1}}{K}) \\
&= \overline{\beta}_{\tau+1} + \frac{\overline{\alpha}_{\tau+1}}{K} + \frac{\overline{\alpha}_\tau[(1 - \alpha_{\tau+1}) - 1]}{K} \\
&= \overline{\beta}_{\tau+1};
\end{aligned}
\tag{21}
$$

4. When $x \neq x_0$ and $\operatorname{argmax} x_0 \in [K, K + K^*)$:

$$
\begin{aligned}
Q_{\tau+1}x_0 &= \overline{\beta^*}_\tau(\alpha_{\tau+1} + \beta^*_{\tau+1}) + \overline{\beta^*}_\tau \beta^*_{\tau+1}(K^* - 1) + \overline{\alpha}_\tau \beta^*_{\tau+1} \\
&= \overline{\beta^*}_\tau(\alpha_{\tau+1} + \beta^*_{\tau+1}) * K^* + \overline{\alpha}_\tau \beta^*_{\tau+1} \\
&= \frac{1 - \overline{\alpha}_\tau - \overline{\gamma}_\tau}{K^*} * (1 - \gamma_{\tau+1}) + \overline{\alpha}_\tau \beta^*_{\tau+1} \\
&= \frac{1}{K^*}(1 - \overline{\gamma}_{\tau+1}) + \overline{\alpha}_\tau(\beta^*_{\tau+1} - \frac{1 - \gamma_{\tau+1}}{K^*}) \\
&= \overline{\beta^*}_{\tau+1} + \frac{\overline{\alpha}_{\tau+1}}{K^*} + \frac{\overline{\alpha}_\tau[(1 - \alpha_{\tau+1}) - 1]}{K^*} \\
&= \overline{\beta^*}_{\tau+1};
\end{aligned}
\tag{22}
$$

5. When $\operatorname{argmax} \boldsymbol{x} = K + K^*$:

$$
\begin{aligned}
\boldsymbol{Q}_{\tau+1}\boldsymbol{x}_0 &= \overline{\gamma}_\tau + (1 - \overline{\gamma}_\tau)\gamma_{\tau+1} \\
&= 1 - (1 - \overline{\gamma}_{\tau+1}) \\
&= \overline{\gamma}_{\tau+1}.
\end{aligned} \tag{23}
$$

Thus, we can obtain a closed-form solution for $\overline{\boldsymbol{Q}}_t\boldsymbol{x}_0$:

$$
\overline{\boldsymbol{Q}}_t\boldsymbol{x}_0 = \begin{cases} (\overline{\alpha}_\tau + \overline{\beta}_\tau)\boldsymbol{x}_0 + (\overline{\gamma}_\tau - \overline{\beta}_\tau)\boldsymbol{x}_{[\mathrm{M}]} + \overline{\beta}_\tau, & \mathbf{s.t.}\,\operatorname{argmax}\boldsymbol{x}_0 \in [0, K), \\ (\overline{\alpha}_\tau + \overline{\beta^*}_\tau)\boldsymbol{x}_0 + (\overline{\gamma}_\tau - \overline{\beta^*}_\tau)\boldsymbol{x}_{[\mathrm{M}]} + \overline{\beta^*}_\tau, & \mathbf{s.t.}\,\operatorname{argmax}\boldsymbol{x}_0 \in [K, K + K^*). \end{cases}' \tag{24}
$$
where $\boldsymbol{x}_{[\mathrm{M}]} = \boldsymbol{x} \leftarrow \operatorname{argmax}\boldsymbol{x} = K + K^*$.

Given indicator functions $\mathbb{1}$ and $\mathbb{1}^*$ for different modals:

$$
\mathbb{1}(\boldsymbol{x}_0) = \begin{cases} 1 & \text{if } \operatorname{argmax}\boldsymbol{x}_0 \in [0, K), \\ 0 & \text{otherwise}. \end{cases}
$$
$$
\text{and } \mathbb{1}^*(\boldsymbol{x}_0) = \begin{cases} 1 & \text{if } \operatorname{argmax}\boldsymbol{x}_0 \in [K, K + K^*), \\ 0 & \text{otherwise}. \end{cases} \tag{25}
$$
$$
\mathbf{s.t.}\,\mathbb{1}(\boldsymbol{x}_0) \cap \mathbb{1}^*(\boldsymbol{x}_0) = \varnothing,
$$

Eq. 24 can be expressed in:

$$
\overline{\boldsymbol{Q}}_t\boldsymbol{x}_0 = \overline{\alpha}_t\boldsymbol{x}_0 + \left[\overline{\gamma}_t - \mathbb{1}(\boldsymbol{x}_0)\overline{\beta}_t - \mathbb{1}^*(\boldsymbol{x}_0)\overline{\beta}_t^*\right]\boldsymbol{x}_{[\mathrm{M}]} + \mathbb{1}(\boldsymbol{x}_0)\overline{\beta}_t + \mathbb{1}^*(\boldsymbol{x}_0)\overline{\beta}_t^*. \tag{26}
$$

∎

## F    EXTENSION TO MORE MODALITIES

The above description emphasizes textual and visual modalities. However, our proposed methodology is also applicable to other discretisable modalities. Even within the visual domain, we may further consider distinct modalities such as bounding boxes, segmentation masks and edge maps, in addition to RGB images. Specifically, our transition matrix in Eq. 8 is extensible, we can have additional modalities by simply adding new modal quadrants and leaving the mask transition in the final row and column intact. Should the number of states in a modality become excessive, it will degrade to an absorbing diffusion. It is straightforward that the extended unified transition matrix also meets the criteria of the discrete diffusion model, with the stationary distribution assigning all probability mass to the [MASK] token.

The unified objective can also include more modalities. When there are more modalities, $p_\theta(\boldsymbol{x}_{t-1})$ consists of more distributions, e.g., $p_\theta(\boldsymbol{x}_{t-1}^0), p_\theta(\boldsymbol{x}_{t-1}^1), \cdots, p_\theta(\boldsymbol{x}_{t-1}^n)$. The architecture of the neural network used for prediction $\tilde{x}_0$ can also be altered by adding the mutual attention to corresponding modal, which is conditioned on the remaining modal sequences. Due to the fact that the complexity of the transformer is quadratic to the length of the sequence, excessive modalities may incur enormous memory cost.

The above demonstrates the ability of the proposed model to handle complex modalities; however we leave the exploration and optimisation of more modalities to future work.

## G    ADDITIONAL RESULTS

### G.1    IMAGE CAPTION

We provide some image caption results based on CUB-200 and MSCOCO dataset in Figs 6 & 7, respectively.

Ref. Image

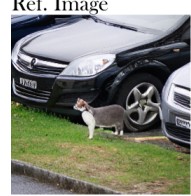

Ref. Caption:
a cat in between two cars in a parking lot .

Sample:
a cat stands in front of a black car on a green car .

Ref. Image

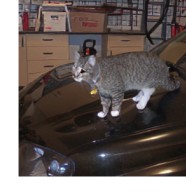

Ref. Caption:
a cat sitting on the hood of a parked black car in a garage .

Sample:
a close up of cat laying on top of a car.

Ref. Image

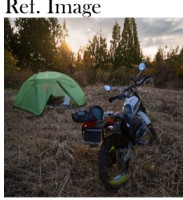

Ref. Caption:
a dirt bike parked near a tent in the woods .

Sample:
a motorcycle parked in a brown grass field .

Ref. Image

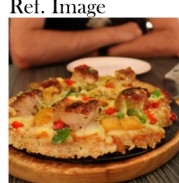

Ref. Caption:
a dish of some sort on a pizza stone .

Sample:
a person is ready for eating a piece of pizza on a plate

Ref. Image

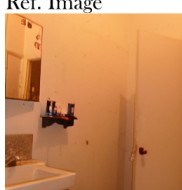

Ref. Caption:
a black trash bag in a restroom next to a sink .

Sample:
a bath tub in a white bathroom sink .

Ref. Image

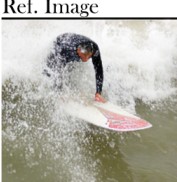

Ref. Caption:
a man in a wetsuit surfing a wave on a surf board .

Sample:
a man is riding a small wave on a surfboard .

Ref. Image

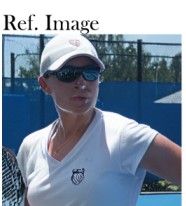

Ref. Caption:
a person on a tennis court in tennis wear .

Sample:
a person standing on a tennis court is about to serve .

Ref. Image

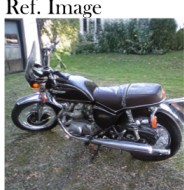

Ref. Caption:
a motorcycle parked on the gravel in front of a garage .

Sample:
a blue motorcycle parked in front of a garage .

Figure 6: Samples of the image caption task based on MSCOCO dataset.

Ref. Image:

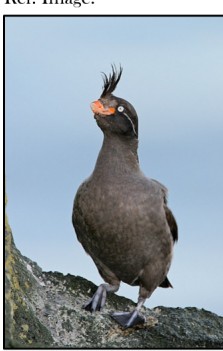

Ref. Captions:

this bird has an all black body with a large orange beak and a white eye.

the bird has a white eyering and webbed black feet.

a bird with a snubbed, rounded orange bill, stark white eyes, webbed feet, and feathered superciliary.

compared to the rest of the body, the head is relatively small, the bird has webbed feet and an orange beak.

this peculiar bird is large and mostly grey, with purple webbed feet, a bright orange, stubby beak, and a black plume on its head.

this bird is black with red and has a very short beak.

a medium sized bird with grey feathers, a black head with a white stripe leading down from the eye, a white eye, a mohawk-shaped feather on the head, and an orange beak.

the crown of the bird has a distinctive feature of black feathers protruding from the head.

this is a dark gray bird with black webbed feet, red beak and a crown that stands up

this bird with webbed feet has an orange bill, a white malar stripe, and a grayish-brown breast.

Samples:

the bird has black feathers on its cheek neck and webbed feet .

this is a dark grey bird with some dark grey wings and a large light colored beak

this big grey bird has a black body and an orange bill .

this is a black and white bird with a long , orange bill .

this ation bird has a white body , short , black beak , grey and black weand orange feet .

this bird has an orange beak with a black body gray belly and black wings

this bird has a grey body and black wings with a long orange bill .

this bird has a dark gray belly , an off orange beak and a bright orange on its eye .

Ref. Image:

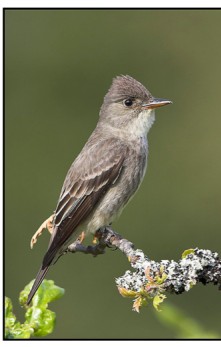

Ref. Captions:

the small bird is light brown with a short beak.

this bird has a white throat, a short bill, and a gray crown.

this small bird with flat bill and a multicolored body , and black eyes.

this small bird is grey and has a small black beak

this is a gray bird with a white throat and a mohawk at the crown.

this particular bird has a gray belly and breast with black secondaries.

this thin bird has a little gray mohawk and a short tail.

this bird is grey with white and has a long, pointy beak.

this particular bird has a belly that is gray with black spots.

this bird has a brown crown with brown coverts and black beak.

Samples:

this is a small brown bird with dark brown wings grey head , and grey breast .

a small brown head and grey bird with a white belly .

small bird with brown feathers and grey back and wings

this bird has wings that are grey and white with a grey belly

this is small gray bird with an all white belly and medium colored wings with a black head .

a small bird with a small bill and a light dark gray belly and it has brown feet .

this small bird is grey and has white wings and tail , and dark gray feathers .

a small grey bird with grey darker on its wings .

Figure 7: More samples of the image caption task based on CUB-200 dataset.

### G.2   Text Conditional Generation

We provide more results of pure text to image synthesis in Fig. 8. The resolution of each generated image is $256 \times 256$.

### G.3   Cross Modal Modification

We present samples for modification across vision and language modals under CUB-200 dataset in Fig. 9. The ref. image shows the reference image after the dVAE reconstruction.

### G.4   Vision-Language Pair Generation

A few additional examples for vision-language pair generation are shown in Fig 10. The resolution of each image is $256 \times 256$.

Text Promotion: this bird has wings that are brown and has a yellow bill .

Image Samples:

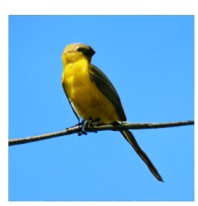 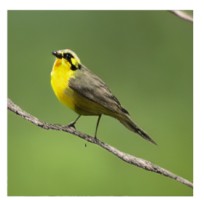 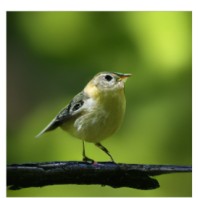 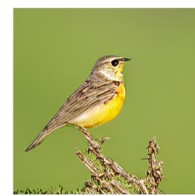

Text Promotion: a completely black bird, small body and head, with a black beak.

Image Samples:

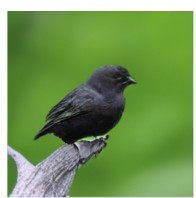 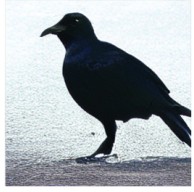 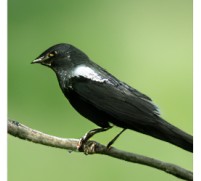 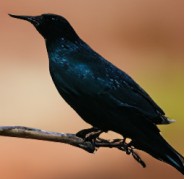

Text Promotion: this particular bird has a belly that is tan and white .

Image Samples:

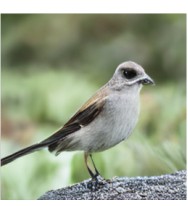 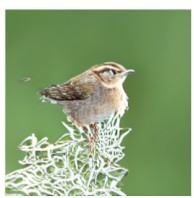 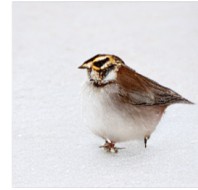 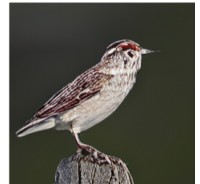

Text Promotion: a parade of motorcycles is going through a group of tall trees .

Image Samples:

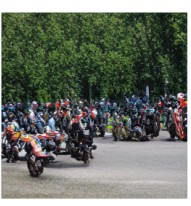 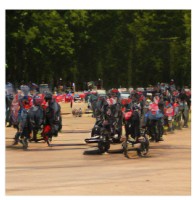 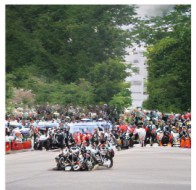 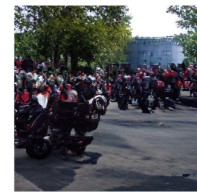

Text Promotion: a city street with multiple bildings and a street light .

Image Samples:

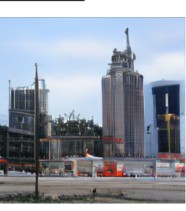 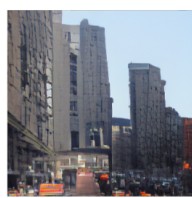 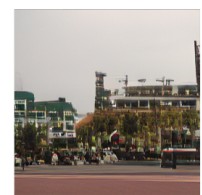 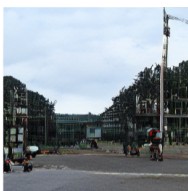

Text Promotion: an all white kitchen with an electric stovetop .

Image Samples:

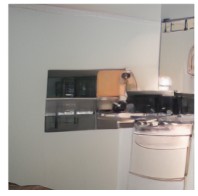 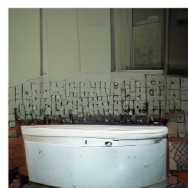 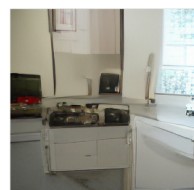 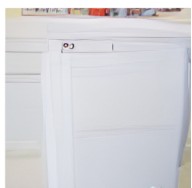

Figure 8: More samples for the pure text-to-image generation from CUB-200 and MSCOCO.

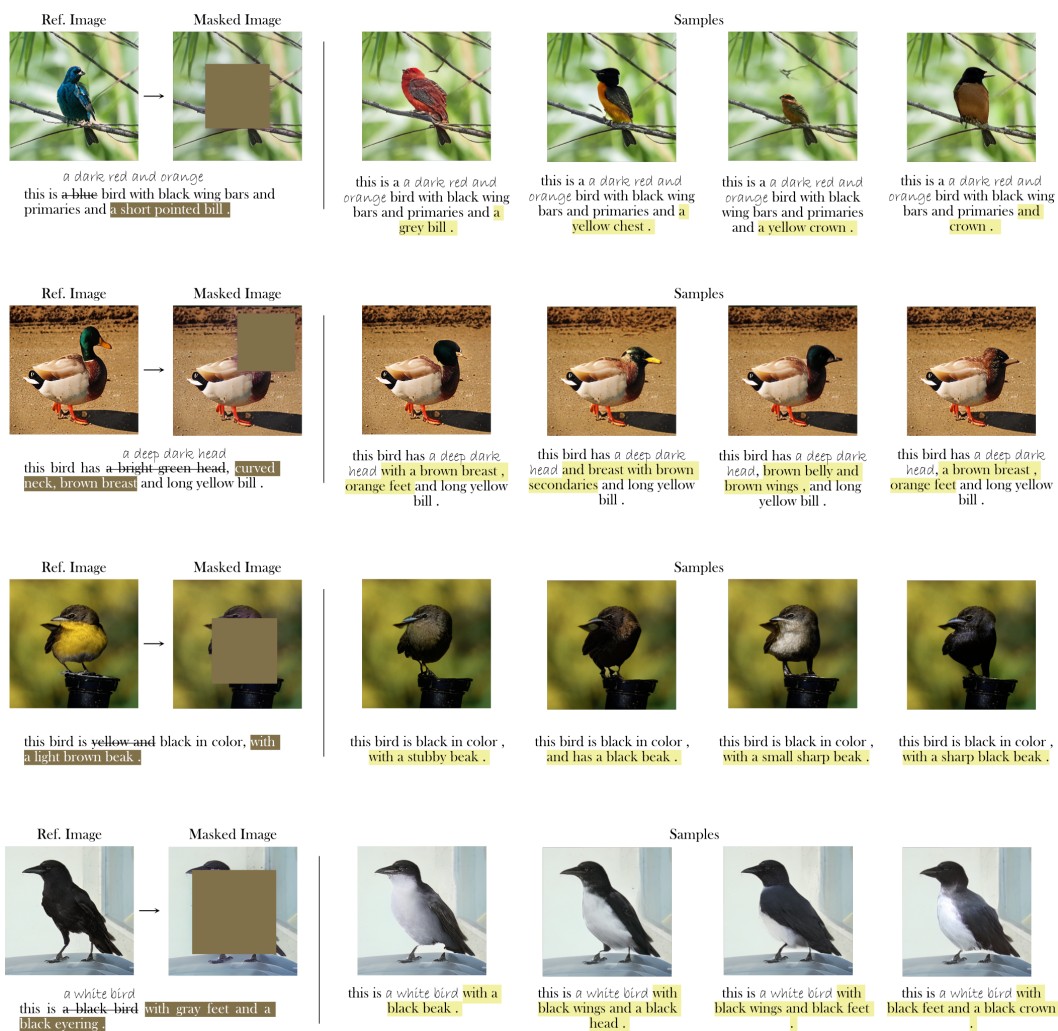

Figure 9: More samples of cross modal vision and language manipulation and infilling. The dark brown portions of the image and description represent the [MASK], while the strike-through represent the caption manipulation.

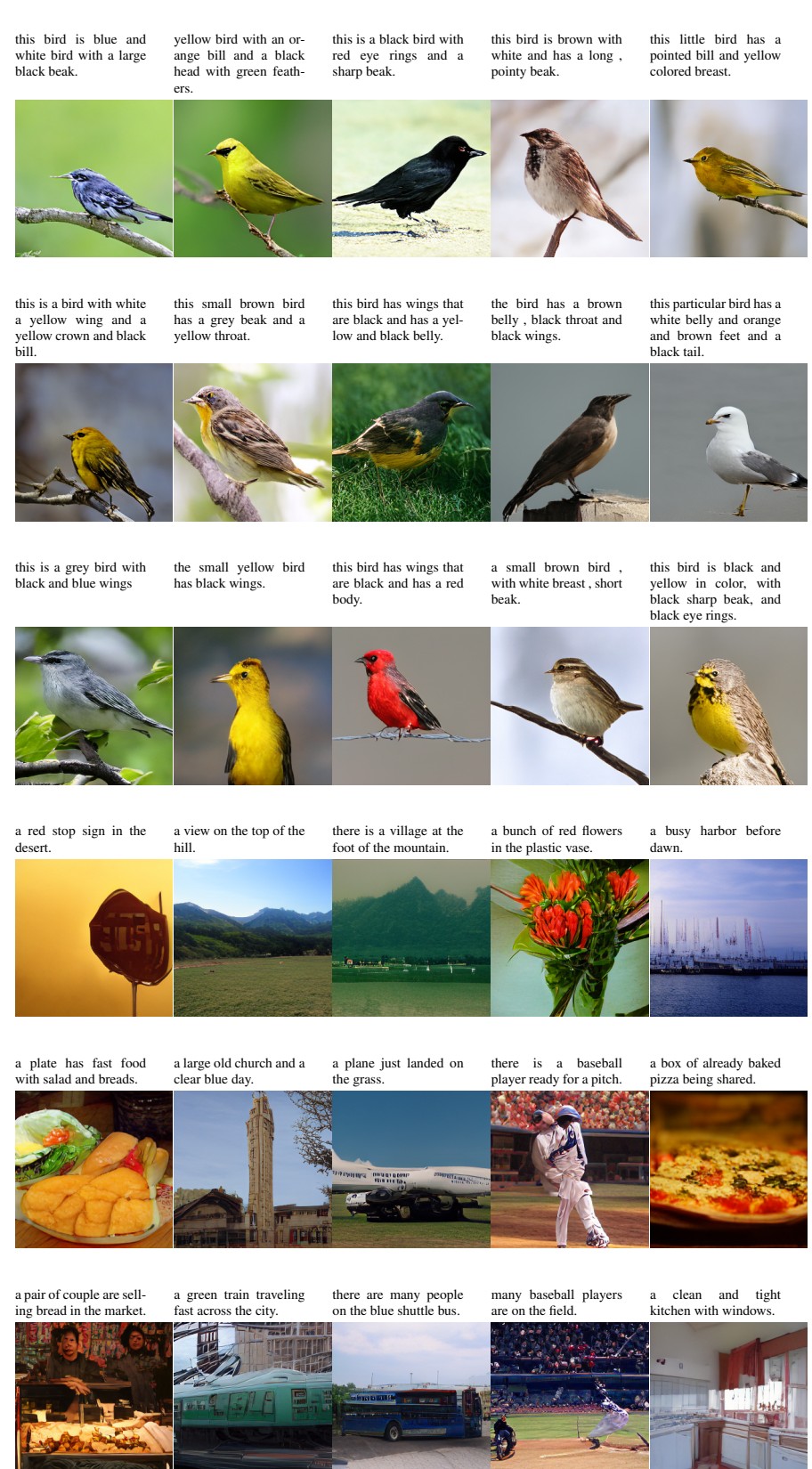

Figure 10: More vision-language pair samples from CUB-200 and MSCOCO. Both the image and caption are *generated simultaneously.*

