# OpenReview forum: "Unified Discrete Diffusion for Simultaneous Vision-Language Generation"
_ICLR.cc/2023/Conference — ICLR 2023 poster_

### Official Review · Reviewer_1mAH · 2022-10-22

**Confidence:** 3
**Clarity, Quality, Novelty And Reproducibility:** This paper is clear. While the techni…
**Correctness:** 2
**Technical Novelty And Significance:** 2
**Empirical Novelty And Significance:** 3
**Recommendation:** 5

**Strength And Weaknesses:**

Strength:

The experiments are interesting for me. For example, it can mask parts of images and captions, and then generate the masked parts, which might be useful in practice.

Questions:

1. Missing details of conditional generation. This paper models a joint distribution $p(x^{img}, x^{txt})$, where sampling from the joint distribution is obvious. However, it is not obvious for me to sample from the condition distribution, such as $p(x^{img} | x^{txt})$. How does this work implement it?

2. Missing details of image impainting. The mask is drawn in the pixel space, instead of the token space. So the implementation of impainting is not so obvious. I suggest to add these details in Appendix. Besides, does this method support mask an arbitrary area in the image? How to ensure the consistency in the unmasked part between the original image and the generated image?

3. In Eq.(10), why the conditional distribution is $p_\theta(x_0^{img} |x_1, x_0^{txt} )$ instead of $p_\theta(x_0^{img}|x_1)$? I think $p_\theta(x_0|x_1) = p_\theta(x_0^{img}|x_1) p_\theta(x_0^{txt}|x_1)$, and therefore $L_0$ should be $-E_q [\log p_\theta(x_0^{img}|x_1) + \log p_\theta(x_0^{txt}|x_1)]$.

4. In Eq.(12) and Eq.(13), the left part is unrelated to $x_0^{txt}$, but the right part is related to $x_0^{txt}$. Is this a typo?

5. This work uses a large number of parameters (600M) for relatively small dataset (MSCOCO), but there is still a performance gap between it and the compared method, e.g., FID=25.11 v.s. FID=10.5 on MSCOCO. Perhaps there is still an improvement space on the architecture.

Summary:

Overall, I think this paper is interesting, but it can still be improved by adding more details and fixing some confusions.

**Summary Of The Paper:**

This paper models the joint distribution of texts and images using a discrete diffusion model. The basic idea is to firstly using a VQ model to convert an image to discrete tokens, and then model tokens of both texts and images. This paper designs a specific Markov transition matrix as well as a mutual attention mechanism for this joint modeling task. By joint modeling, this paper can do tasks such as joint sampling of texts and images, text to images, and image captioning.


**Summary Of The Review:**

See Strength and Weaknesses.

---

> ### Author Response · Authors · 2022-11-10
> **Reply to Reviewer 1mAH**
>
> ### Q1 : How does this work implement the conditional generation?
>
> The sampling process of diffusion models can be regarded as the Markov chain.  The model is trained to predict $p(x_{t-1}|x_{t})$, while $t \in [0, T]$, $p(x_0)$ is the clear image and $p(x_T)$ is the noisy pattern. In our case, once our model learnt $p([x_{t-1}^{img}, x_{t-1}^{txt}] | [x_{t}^{img}, x_{t}^{txt}])$, the conditional generation part is straightforward, $x_t^{img}$ and $x_t^{txt}$ can be either information or noisy tokens. Generally speaking, our model is still leaning $p(x_{t-1}|x_t)$, where $x_t$ consists of given condition, [MASK]s and diffused tokens.
>
> ### Q2 : Missing details of image impainting.
>
> - How does the image inpainting implement?
> We follow the widely used inpainting methods for the VQ / Discrete model, there are some publications [Esser et al. (2021), Gu et al. (2022)] for reference. We simply set the tokens in the irregular masked region as [MASK] tokens and fed them to our generative model.
> - Does this method support mask an arbitrary area in the image?
> Yes.
> - How to ensure the consistency in the unmasked part between the original image and the generated image?
> The VQ model is geographically relevant in that changes to local tokens have a negligible effect on the reconstruction results of distant tokens. In another words, the mask will affect the embedded tokens in the masked area severely but influence the unmasked part slightly. However, depending on the power of the VQ-model, the details in the raw image could be lost during recovery.
>
> ### Q3 : Why the conditional distribution is $p_{\theta}(x_0^{img}|x_1, x_0^{txt})$?
>
> Mathematically, $p(x_0^{img})$ not only condition on $p(x_1)$, as $p(x^{img}_t)$ and $p(x^{txt}_t)$ also has some inter connections. We would like to highlight the relationship between different modalities at the same time-step t as well as the connections between the modalities in the current state t and the previous state t+1.
>
> Practically, we proposed the mutual attention module, which enable the implementation of $p_{\\theta}(x_{t-1}^{img}|x_t, \\widetilde{x_0^{img}}, x_0^{txt})$ and  $p_{\\theta}(x_{t-1}^{txt}|x_t, \\widetilde{x_0^{txt}}, x_0^{img})$ The detail of mutual attention can be found in the manuscript, and we also provide a more clear explanation about it in the Reply to Reviewer bEZ6.
>
> ### Q4 : Are Eqs 13 and 14 typo as left side is unrelated to text modal?
>
> It’s not a typo. As we discussed in Q3, the current modal both condition on the previous whole state and the current state of all other modals.
>
> ### Q5 : There is still a performance gap between our work and other compared methods.
>
> As several reviewers raised the same concerns about our model performance, please allow me to repeat some of my responses to other reviewers.
>
> - Our model solves the multi-modal generation problems in a totally novel way. We achieve the unifying of multiple modalities' modelling with a single model. In contrast to the approaches for comparison, in which all models can only generate the content of ONE modality based on the given condition of one or more modalities, our method is able to generate the content of several modalities concurrently without the requirement for a given condition. This power to generate is unique.
> - It is therefore somewhat unfair to use the performance of the modality conversion model to measure the performance of our modality generation model.
> - The detailed discussion between our methods and VQ-Diff, Autoregression Model can be found in the reply to Reviewer cr8c and bEZ6 separately.
>
> However, we acknowledge that there is still much work to be done on the structure of the model or the details of the pipeline to improve the performance.
>
> About the model size, we also have a model for CUB with only 117M params, whose hidden dim is 256 instead of 1024. The FID is 19.14 (vs 17.38 mentioned in the paper). At this stage, the impact of the number of model parameters is not somewhat important.
>
> ### Finally, thank you for your comments on our manuscript. We will consider presenting more related literatures and explanations in more detail in our manuscript to make our work more accessible for general readers.

---

> > ### Comment · Reviewer_1mAH · 2022-11-21
> > **Further questions**
> >
> > Thanks for the reply. Here are further questions.
> >
> > Q1: Can you specify how to determine $x_t^{txt}$ when you want to sample a image conditioned on a given text $x^{txt}$?
> >
> > Q2: Can you specify details of impainting when the mask area is arbitrary?
> >
> > Q3: According to the reply, the author should implement $p_\theta(x_0^{img}|x_1, x_0^{txt})$. It requires a neural network that takes three inputs: $x_1^{txt}, x_1^{img}, x_0^{txt}$. However, in Section 3.2, the proposed neural network only adopts two inputs $x_t^{txt}, x_t^{img}$.
> >
> > Q4: The author still does not explain the mathematical contradiction that the left part is unrelated to $x_0^{txt}$ and the right part is related to $x_0^{txt}$ in Eq.(12).

---

> > > ### Author Response · Authors · 2022-11-21
> > > **Reply to further questions from Reviewer 1mAH**
> > >
> > > Q1. The forward process can be deterministic and closed form solved. Once we have the condition, we can keep the conditional signal fixed.
> > >
> > > Q2.  Following the existing image VQ methods, given an image degraded by a random mask with an arbitrary area, the masked regions will be automatically set to zero. The quantizer then directly embeds the masked image into the discrete latent space with the discrete index. However, the decoded outputs for the irregularly masked input will be slightly different, even for the originally visible region. We can further use the $I_{out}=(1-mask) * I_{out} + mask * I_{in}$ to get the final output and maintain the original visible contents. This is maybe a key issue for vq-based image inpainting task (which will apply another refinement network to ensure the consistency between the generated mask regions and the original visible regions), but it is not the key consideration in our case.
> > >
> > > Q3.
> > > * Estimating $p_{\theta}(x_0^{img}|x_1, x_0^{txt})$ does not require a neural network.
> > >
> > > * The input of neural networks is $x_t$ or as the reviewer mentioned $x_t^{img}$ and $x_t^{txt}$.
> > >
> > > * The loss function for t>0 consists of different modalities, $p_{\\theta}(x_{t-1}^{img}|x_t, \\widetilde{x_0^{img}}, x_0^{txt})$ and $p_{\theta}(x_{t-1}^{txt}|x_t, \\widetilde{x_0^{txt}}, x_0^{img})$, where $\\widetilde{x_0}$ are the re-parameterisations and the only part to be predicted by the neural networks. Therefore, the required inputs for the neural networks to predict $\tilde{x}_{0}$ in the training phase are just $x_t$.
> > >
> > > Q4. The discrete diffusion is not exactly the same as the Gaussian diffusion. We need to estimate the $\\widetilde{x_0}$ as the re-parameterisation, which not only contains the modality to be estimated, but also has the given modality as ground truth in the training phase. And the answer in Q3 also provides some hints for this question, the neural network is responsible for prediction of the distribution $\\tilde{p_{\theta}}(\\widetilde{x_0}|x_t)$. However, the input to the neural network covers all modalities, and a simple self-attention mechanism can scarcely highlight the inter-modal linkages. In addition, the distributions predicted by the neural network need to be decoupled according to the various modalities throughout the reparameterisation. In other words, the network predictions should be expressed in terms of different modalities, e.g., $\tilde{p_{\theta}}(\\widetilde{x_0^{img}}|x_t)$ and $\tilde{p_{\theta}}(\\widetilde{x_0^{txt}}|x_t)$, with interconnections.

---

> > > ### Author Response · Authors · 2022-11-30
> > > **Further discussion will be highly appreciated.**
> > >
> > > Dear reviewer 1mAH,
> > >
> > > We would like to thank you again for your futher queries. In our responses, we have made every effort to answer your queries about the intra- and inter- relationship between different modalities. We have also attempted to clarify the equations and their significance. We eagerly await your further response to our replies, and we welcome any suggestions for improving our effort.
> > >
> > >
> > > Best.

---

### Official Review · Reviewer_tuqJ · 2022-10-23

**Confidence:** 3
**Correctness:** 4
**Technical Novelty And Significance:** 4
**Empirical Novelty And Significance:** 3
**Recommendation:** 8

**Clarity, Quality, Novelty And Reproducibility:**

The paper needs some improvement in regards to clarity.

The paper is novel and the method is of broad interest.

The paper is accompanied by code.

**Strength And Weaknesses:**

The paper is conceptually easy to follow and understand, and has the merit to propose a unified model for both image to text, and text to image synthesis as well as completion. The paper is reproducible and the computational requirements to achieve good results are not excessive.

The quantitative results do not seem to surpass previous works on either of the modalities, which might at first set some concerns with the proposed work. However, it is worth noting that the metrics do not fully reflect which model is better, and the visual results are really on par with existing methods (For example the FID score is a metric that below some level is indeed indicating a dataset replication). It is also worth remarking the merit of the method to achieve competitive results in both tasks with the same model. In addition, the multi-modal completion feature is a desirable property of the method which to my knowledge has not been proposed yet in the literature.

While the paper is technically sound and easy to follow, it reads quite weak; the writing and presentation need a good effort before the camera ready to improve the paper’s readability. There are several typos and while the mathematical derivation is interesting, some other details might be more attractive to the audience that are left to the supplementary material, such as the experimental details. I would thus suggest the authors to perform a thorough proofreading.

Some well renowned methods are left out of the comparisons, such as DALL-E [Ramesh et al. “Zero-Shot Text-to-Image Generation”], for the task of image generation. While DALL-E is a very big model, showing how the model’s capacity can affect performance would also boost the paper’s contributions.


**Summary Of The Paper:**

The paper presents a method for image-text generation that can perform conditional and unconditional image and text synthesis from either blank inputs or by partially masking any of them. The method includes a two pathway for image and text encoding-decoding with a unified diffusion path and matrix, that works with the fusion of the visual and text tokens. The paper is accompanied with code. The experiments include text-to-image synthesis, image captioning, and image and text replacement, which can be accomplished by the very same model. The results are competitive w.r.t. state of the art works, setting a promising direction towards multimodal generation.

**Summary Of The Review:**

The paper has the merits to be published at ICLR, and it has the potential to draw the audience's eye. However, some extra efforts are needed towards improving the quality of the manuscript.

---

> ### Author Response · Authors · 2022-11-10
> **Reply to Reviewer tuqJ**
>
> ### We appreciate your awareness of the novelty of our work, which is a great encouragement to us.
>
> ### Q1 : Why DALL-E was not included in the comparision?
>
> DALLE was very influential in the field of multimodal generation based on discrete representations, and our work is also inspired by DALLE. We make many references to this work throughout the manuscript.
> However the DALLE numerical results are all based on ZERO shot generation and are not fine-tuned on CUB or MSCOCO. To make a simple comparison:
>
> | FID, lower is better. | CUB | MSCOCO |
> | --- | --- | --- |
> | DALL.E | 28.3 | 27.5 |
> | Uni3D | 16.19 | 25.11 |
>
> ### Q2 : How does the model scale influence the performance?
>
> During the pre-experimental phase, we roughly test our model performance for image generation on CUB dataset. We choose the dim of 256 (117M) and 1024 (600M), the results are shown below:
>
> |  | Dim 256 | Dim 1024 |
> | --- | --- | --- |
> | FID | 19.14 | 17.38 |
>
> ### Q3 :  The paper reads quite weak and the writing and presentation need a good effort.
>
> In this response phase, we present some new concepts to make our point stronger, i.e., “Modal Translation” and “Modal Generation” in Summary Reply. We consider incorporating the relevant statements into a revised manuscript.
>
> We will carefully proofread the manuscript after adding relevant content.
>
> ### Misc.
> Additionally, we are also interested in proposing a metric to evaluate multi modality generative tasks,  including the sample quality, the connections between different modalities, etc. Improving model details to enhance model performance is a also crucial aspect of future work.
>
> ### Last but not least, thanks for the recognition of our work.

---

### Official Review · Reviewer_bEZ6 · 2022-10-24

**Confidence:** 4
**Correctness:** 3
**Technical Novelty And Significance:** 3
**Empirical Novelty And Significance:** 3
**Recommendation:** 6

**Clarity, Quality, Novelty And Reproducibility:**

- The paper is not clearly written, e.g. the introduction of mutual attention.
- Novelty is a little limited given that the unified transition matrix is a trivial and intuitive extension of multinomial diffusion to the joint text-image sequence.
- Code is provided in Supp.

**Strength And Weaknesses:**

## strengths
- The proposed method is simple and shows promising results.
- The proposed unified framework supports various tasks including cross-modal, text-to-image, and image-to-text generation.
## weakness
- The paper is not clearly written. For example, I cannot fully understand the proposed mutual attention from Fig. 3. It would be good if the authors could provide e.g. mathematical definitions.
- One of the main contributions: the unified transition matrix seems a trivial and intuitive extension of VQ-Diffusion or multinomial diffusion to the concatenated text-image tokens. The novelty is slightly limited.
- Autoregressive models like DALLE also model the joint distribution of text and image. How does the proposed model compare to autoregressive models?
- From the results shown in Tables 1 & 2, the proposed method does not perform the best in any of the entries. Could the author provide more discussion on the results? For example in Table 1, the T2I FID is much higher than VQ-Diffusion, which is a very close baseline. Any intuition why joint generative vision-language training is not helpful here?

**Summary Of The Paper:**

The paper proposes a unified discrete diffusion model for simultaneous vision-language generation. The proposed model extends multinomial diffusion to the joint text-image tokens with a transition matrix to prevent transiting among modalities. A mutual attention module is proposed to better capture the inter-modal linkages.

**Summary Of The Review:**

My rating is mainly based on the novelty and clarity of the paper. I might amend my score if my major concerns are addressed.

---

> ### Author Response · Authors · 2022-11-10
> **Reply to Reviewer bEZ6 (1/3)**
>
> ### Q1: Some concepts in the paper are not clear, e.g. the mutual attention.
>
> Our mutual attention is a variant of cross attention, which is widely used in multi-modal tasks. Cross-attention is able to integrate asymmetrically two distinct embedding sequences of the same dimension, one of which is the Query from the modality to be generated $X$ and the other are the Key and Value with the conditional modality $C$. In the conventional multi-model generative model,  conditional modality is given and fixed. Thus the Cross attention block can be expressed by:
>
> $$
> \text{CA}(X,C) = Attn(X,C;W) = \text{softmax} (\frac{QK^T}{\sqrt{2}})V,
> $$
>
> where $Q=W_QX, K=W_KC\\  \\& \\ V=W_VC$ , and $W$ is the learnable parameters. Noted that the cross attention $CA(X,C)$ could convert to self-attention $SA(X)$ when $X = C$.
>
> In our model, each modality may become a component of what needs to be generated. Therefore, we propose mutual attention to enable tokens of different modalities in a sequence to be conditional on each other, allowing the capture of the relationships between the various modalities. Mathematically, our mutual attention can be expressed as
>
> $$
> \text{MA}(T_i,T_j) = Attn(T_{i}, T_{j}; W) = \text{softmax} (\frac{Q_iK_j^T}{\sqrt{2}})V_j,
> $$
>
> where $Q_i=W_QT_i, K=W_KT_j\\  \\& \\ V=W_VT_j$ , and $T_i \\ \\& \\ T_j$ are the tokens in a different modality, e.g. image and text. The Unified block contains a self-attention and several mutual attention in parallel. Given a hidden token vector $H' = [H'_i, H'_j]$, the whole pipeline of Unified block is
>
> $$
> \begin{aligned}
> 1.&T' = SA(H') \\\\
> 2.&\text{Decouple:}\ T' \\rightarrow T'_i, T'_j \\\\
> 3.&\begin{cases}
> T_i = \text{MA}(T'_i, T'_j) \\\\
> T_j = \text{MA}(T'_j, T'_i) \\\\
> \end{cases} \\\\
> 4.&\text{Couple:}\ T \leftarrow T_i, T_j \\\\
> \end{aligned}
> $$
>
> The above equations for $\text{MA}$ will be embedded in the revised manuscript. If there is still confusion about the mutual attention or other parts of our work, welcome to point them out.
>
> ### Q2 : The novelty of the Unified Transition Matrix (UTM) is slightly limited.
>
> We propose the discrete diffusion model with UTM to ensure that the tokens of the different modalities remain independent during the diffusion process.  In contrast to the more popular Gaussian diffusion model (or other diffusion models in the continuous domain), the discrete diffusion has an editable transition matrix, such a property was mentioned in D3PM [1]. VQ-Diffusion [2] extends the multinomial diffusion to a mask type, or called absorbing diffusion in D3PM, which is inspired by the masked language model. Both of them did not assign a specific target to the transition matrix modification. Our work is the first to design a unique transfer matrix to achieve a purposeful diffusion process to the best of our knowledge.
>
> We believe this approach can be instructive for the community, especially for applications of discrete diffusion models. We can achieve a controlled diffusion process by designing unique transfer matrices for modalities such as text, images, etc., and thus have more precise control over the sampling results.
>
> We will mention this in the Introduction and Conclusion part of the manuscript and hope to inspire the reader to think about this aspect of designing transfer matrices for the discrete diffusion models.

---

> > ### Author Response · Authors · 2022-11-10
> > **Reply to Reviewer bEZ6 (2/3)**
> >
> > ### Q3 : How does the proposed model compare to autoregressive models?
> >
> > Although it has been demonstrated that autoregressive models based on Transformer such as DALLE[3] and Parti [4] perform well on text-to-image generation tasks, we do not feel that they models the joint distribution of images and text, instead they only estimate a marginal distribution condition on text. The detail can be found in our summary response. It is worth noting that all of such models are “Modality Translation” models, which means they can only achieve the translation from one modal to the other, i.e., Text to Image. The main reason is that AR process must assign an ordering, either directional or bi-directional. Therefore, AR models are able to offer satisfactory results in text-to-image tasks (only), as they can put the text condition tokens at the beginning of the sequence.
> >
> > Additionally, AR models are incapable of capturing global features, whereas diffusion models do well [5]. Thus our model does not take into account the position of the conditional tokens in the sequence; the given conditional tokens can be interspersed with the tokens to be generated. And we can also implement text-to-image generation, image caption, multi-modal manipulation with only one model due to this fact.
> >
> > ### Q4 : Could the author provide more discussion on the results compared with VQ-Diffusion?
> >
> > As your question is very similar to Q2 of the reviewer cr8c, please allow us to copy some of the content to address your concerns. You are also welcome to refer to my response to the reviewer cr8c, where we also discuss the influence of jointly training in detail with experimental results.
> >
> > Despite the fact that our model superficially resembles VQ-Diffusion, the actual implementation and intended goals are fairly different.
> >
> > - VQ-Diffusion intends to solve the Text-to-Image generation problems while there must be a given text condition, which is provided by CLIP in a continuous vector format and injected into the time embedding. In other words, it is permissible to manipulate the conditional space for a better representation, which can alleviate the burden on the generative model to represent the condition. Some work proved that a powerful text encoder (T5 [6]) and a highly abstract text representation could provide a better result [7].
> > - Our work focuses on multi-modal generation, and the task of text generation is also taken into account. We consequently cannot employ adequate text or visual representations that are difficult to recover. Thus putting some pressure on the generative model in order to comprehend and represent the original data.
> > - As the model estimate totally different distributions, i.e., $p(x_{img}|x_{txt})$ in VQ-Diffusion and $p(x_{img},x_{txt})$ in our model, the simple comparison of FIDs does not show which model is superior and VQ-Diffusion cannot be simply regarded as the close baseline of our work.

---

> > > ### Author Response · Authors · 2022-11-10
> > > **Reply to Reviewer bEZ6 (3/3)**
> > >
> > > ### Q5 : Any intuition why joint generative vision-language training is not helpful here?
> > > We disagree somewhat with the assertion that  joint generative vision-language training is not helpful here. As the model estimate totally different distributions, i.e., $p(x_{img}|x_{txt})$ in VQ-Diffusion and $p(x_{img},x_{txt})$ in our model, the simple comparison of FIDs does not show which model is superior.
> > >
> > > The joint generative V-L training provides a more compact connection between generated image and text, which can be found in Table 3. Our model demonstrates sufficient superiority in terms of text-image similarity without optimized by the CLIP loss. In contrast, both improved VQ-Diffusion and OFA make explicitly or implicitly use of CLIP loss to optimize the model parameters.
> > >
> > > For a fair comparison, we train our model in a text-to-image pipeline, during the training phase, we offer the text condition and fixed them in our way (without text encoders, only use BPE tokens). In the inference phase, we generated the images with condition and the image-text pairs without conditions. The results are shown in below:
> > >
> > > | FID in CUB | w/ text condition | w/o text condition |
> > > | --- | --- | --- |
> > > | Text-Conditional | 16.58 | 36.42 |
> > > | Uni3D | 16.19 | 17.38 |
> > >
> > > It can be seen that the joint modelling based on our model not only has the capability of multimodal generation, but also brings some improvement to the quality of image generation.
> > >
> > >
> > > Overall, our study offers a fresh solution to the problem of multimodal generation. In some ways, our work is the first model to implement “Multi-Modalities Generation”, despite the fact that it is even difficult to find a measure to evaluate the generation outcomes (In simultaneous generation task, it is hard to quantify the text quality. All current evaluation methods for text generation results need one or several reference sentences or words.).
> > >
> > > ### Finally, thanks for your contribution in the review of our work. Some points will surely make our work more attractive to general readers.
> > >
> > > [1] Austin, Jacob, et al. "Structured denoising diffusion models in discrete state-spaces." *Advances in Neural Information Processing Systems* 34 (2021): 17981-17993.
> > >
> > > [2] Gu, Shuyang, et al. "Vector quantized diffusion model for text-to-image synthesis." *Proceedings of the IEEE/CVF Conference on Computer Vision and Pattern Recognition*. 2022.
> > >
> > > [3] Ramesh, Aditya, et al. "Zero-shot text-to-image generation." *International Conference on Machine Learning. PMLR, 2021.*
> > >
> > > [4] Yu, Jiahui, et al. "Scaling autoregressive models for content-rich text-to-image generation." *arXiv preprint arXiv:2206.10789* (2022).
> > >
> > > [5] Hu, Minghui, et al. "Global Context with Discrete Diffusion in Vector Quantised Modelling for Image Generation." *Proceedings of the IEEE/CVF Conference on Computer Vision and Pattern Recognition*. 2022.
> > >
> > > [6] Raffel, Colin, et al. "Exploring the limits of transfer learning with a unified text-to-text transformer." *J. Mach. Learn. Res.* 21.140 (2020): 1-67.
> > >
> > > [7] Saharia, Chitwan, et al. "Photorealistic Text-to-Image Diffusion Models with Deep Language Understanding." *arXiv preprint arXiv:2205.11487* (2022).

---

> > ### Comment · Reviewer_bEZ6 · 2022-11-27
> > **Response to authors' rebuttal**
> >
> > I thank the authors for their rebuttal and clarification of the mutual attention module. I have a follow-up question: seems that mutual attention employs all pair-wise cross-attentions and fuse them together, will the computation cost grow quadratically as the number of modalities increases?

---

> > > ### Author Response · Authors · 2022-11-28
> > > **Complexity for Multiple Modalities**
> > >
> > > Thanks for the response.
> > >
> > > The computation cost of our $\text{MA}$ grows linearly. The input to our $\text{MA}$ consists of just two components, regardless of the number of modalities: the context to be generated and the other provided modalities.
> > >
> > > E.g., given a hidden state vector $H'$ that has 3 modalities $i,j,k$, the Unified blocks can be expressed as:
> > >
> > > $$
> > > \begin{aligned}
> > > 1.&T' = SA(H') \\\\
> > > 2.&\text{Decouple:}\ T' \\rightarrow T'_i, T'_j, T'_k \\\\
> > > 3.&\begin{cases}
> > > T_i = \text{MA}(T'_i, [T'_j, T'_k]) \\\\
> > > T_j = \text{MA}(T'_j, [T'_i, T'_k]) \\\\
> > > T_k = \text{MA}(T'_k, [T'_i, T'_j]) \\\\
> > > \end{cases} \\\\
> > > 4.&\text{Couple:}\ T \leftarrow T_i, T_j, T_k \\\\
> > > \end{aligned}
> > > $$
> > >
> > > However, we should note that $\text{SA}$ block in the 1st step requires the quadratic computation time along the token length. If we have more modalities, the sequence length of $H'$ would be longer and cause more cost.

---

> > > > ### Comment · Reviewer_bEZ6 · 2022-11-28
> > > > **thanks for the clarification**
> > > >
> > > > I thank the authors for their clarifications. One of my concerns regarding the clarity of mutual attention is resolved, thus I raise my rating to 6. But I would highly suggest the authors include these discussions in future revision(s), and add ablation studies on different variants of mutual/cross attention.

---

### Official Review · Reviewer_cr8c · 2022-10-26

**Confidence:** 4
**Clarity, Quality, Novelty And Reproducibility:** See above
**Correctness:** 4
**Technical Novelty And Significance:** 2
**Empirical Novelty And Significance:** 2
**Recommendation:** 6

**Strength And Weaknesses:**

Strengths:
1. The paper is clearly written and the method is described in detail. The writing, along with the code, makes the paper reproducible.
2. The method is evaluated thoroughly for image generation, text generation and joint image-text generation using both qualitative and quantitative experiments and compared with baseline methods.


Comments:
1. The proposed method is basically an extension of VQ-diffusion for the multimodal case, with a new mutual-attention transformer with a fused embedding proposed to enable multimodal generation. While this extension is not trivial, it is fairly intuitive.  Can the authors highlight the main novelty and technical contributions of the paper?
2. The generation results for individual modalities are competitive, but worse than VQ-diffusion, indicating that the joint distribution estimation leads to degradation in performance for T2I generation.  Can the authors comment on this?
3. The multimodal generation results are significantly worse than the autoregressive approach (OFA). While its a very different approach, it would be good for authors to comment on the comparison.


**Summary Of The Paper:**

This paper proposes a multimodal diffusion model, that can perform text-conditioned, image-conditioned and text-and-image-conditioned generation. They utilize discrete diffusion, based on VQ-diffusion [Gu et al, 2022], with the images encoded using discrete VAE and the text encoded using byte-pair encoding. The  discrete diffusion process is designed in the form a unified text-image Markov transition matrix to estimate the joint distribution of language and image. To implement this in practice, they propose a mutual-attention transformer with a fused embedding, which enables the unified diffusion process. They perform experiments for image generation and text-image generation using the CUB and MSCOCO datasets and compare their method with baselines using FID and IS scores. The proposed method is worse than the baseline VQ-diffusion on both datasets, but obtains competitive FID scores. Image caption generation is also evaluated on MSCOCO, where the proposed approach obtains competitive (but not state-of-the-art) results. However, uniquely, the method is able to simultaneously generate both text-and-image and this capability is evaluated by comparing the CLIP similarity between generated image-caption pairs with CLIP scores of eval set pairs, where the proposed method shows slightly better results.

**Summary Of The Review:**

The paper proposes an extension of VQ-diffusion for multimodal discrete diffusion generative model. The results are reasonably good and the novelty of the approach is not somewhat limited.

---

> ### Author Response · Authors · 2022-11-10
> **Reply to Reviewer cr8c**
>
> ### Q1 : Can the authors highlight the main novelty and technical contributions of the paper?
>
> In addition to the innovations in the novel tasks we have discussed in the Summary Response, our paper mainly has 3 technical contributions to achieve the “Multi-Modalities Generation”
>
> 1. Unified Transition Matrix.
>
>     We propose the discrete diffusion model with the Unified Transition Matrix to control the forward process, and ensure the tokens of the different modalities could fall into the correct domain after diffusion.  In contrast to the more popular Gaussian diffusion model (or other diffusion models in the continuous domain), the discrete diffusion has an editable transition matrix, such a property was mentioned in D3PM [1]. And VQ-Diffusion [2] extends the multinomial diffusion to a mask type, (or called absorbing diffusion in D3PM), which is inspired by the masked language model. Both of them did not assign a specific target to the transition matrix modification. Our work is the first to design a unique transfer matrix to achieve a purposeful diffusion process to the best of our knowledge.
>
>     We believe this approach can be instructive for the community, especially for applications of discrete diffusion models. We can achieve a controlled diffusion process by designing unique transfer matrices for specific modalities such as text, images, etc., and thus have more precise control over the sampling results.
>
> 2. Unified Objective Functions
>
>     In the discrete diffusion model, the neural network is responsible for predicting x0 rather than noise $\epsilon$ . During the training process, we want the model to capture both the interrelationships between different modalities and the inner connections between the same modalities, e.g. $p_{\theta}(x_0^{i}|x_1, x_0^{j})$, where i and j represent different modalities. Therefore, we propose the Unified Objective Function, a modified version based on the Diffusion model, or VPSDE.
>
> 3. Mutual Attention
>
>     Our mutual attention is a variant of cross-attention. As each modality in our work may become a component of what needs to be generated, we use mutual attention to enable tokens of different modalities in a sequence to be conditional on each other, allowing capture of the relationships between the various modalities. The detail of mutual attention can be found in the manuscript, and we also provide a more clear explanation about it in the Reply to Reviewer bEZ6.

---

> > ### Author Response · Authors · 2022-11-10
> > **Reply to Reviewer cr8c continued**
> >
> > ### Q2 : Can the authors comment on the gap between the proposed work and VQ-Diffusion and  joint distribution estimation leading the degradation in performance for T2I generation?
> >
> > Despite the fact that our model superficially resembles VQ-Diffusion, the actual implementation and intended goals are fairly different. As we have mentioned before, VQ-Diffusion is a “Modality Translation” model, while our work focuses on “Multi-modalities Generation”.
> >
> > - VQ-Diffusion intends to solve the Text-to-Image generation problems while there must be a given text condition, which is provided by CLIP in a continuous vector format and injected into the time embedding. In other words, it is permissible to manipulate the conditional space for a better representation, which can alleviate the burden on the generative model to represent the condition. Some work proved that a powerful text encoder (T5 [3]) and a highly abstract text representation could provide a better result [4].
> > - Our work focuses on multi-modal generation, and the task of text generation is also taken into account. We consequently cannot employ adequate text or visual representations that are difficult to recover. Thus putting some pressure on the generative model in order to comprehend and represent the original data.
> > - We disagree somewhat with the assertion that the estimate of the joint distribution degrades the quality of the image-to-text translation. As the model estimate totally different distributions, i.e., $p(x_{img}|x_{txt})$ in VQ-Diffusion and $p(x_{img},x_{txt})$ in our model, the simple comparison of FIDs does not show which model is superior.
> > - For a fair comparison, we train our model in a text-to-image pipeline, during the training phase, we offer the text condition and fixed them in our way (without text encoders, only use BPE tokens). In the inference phase, we generated the images with condition and the image-text pairs without conditions. The results are shown in below:
> >     | FID in CUB | w/ text condition | w/o text condition |
> >     | --- | --- | --- |
> >     | Text-Conditional | 16.58 | 36.42 |
> >     | Uni3D | 16.19 | 17.38 |
> >
> >     It can be seen that the joint modelling based on our model not only has the capability of multimodal generation, but also brings some improvement to the quality of image generation.
> >
> > - Moreover, The joint generative V-L training provides a more compact connection between generated image and text, which can be found in Table 3. The similarity score is better than VQ-Diffusion. Improvements to the model's performance are necessary. We would focus on the FID results in our future work.
> >
> > ### Q3 : Can the authors comment on the gap between the proposed work and OFA?
> >
> > The OFA is a “Multi-Modalities Translation Generative Model”. Instead of giving unimodality as the condition, OFA accepts multiple modalities. For each modality, OFA has specific agents to extract the features. Besides, the OFA needs a pretraining process, the model weights are initialized as the released BART model. The pretraining dataset includes OpenImages, YFCC100M, ImageNet-21K, CC12M, CC3M, SBU, MSCOCO, and VG-Captions. And for each downstream task, e.g., Text-to-Image or Image Caption, OFA needs a finetuning on a specific dataset.
> >
> > As we have mentioned before, such “Modality Translation” models do not consider the capacity to recover the raw content of the given conditional modality. The representation of the conditional modalities is handled by the other pre-trained networks. In addition, as large-scale pre-trained models, the quantity of data required during the pre-training phase affects their final performance. The authors also mentioned the initialization parameters of the OFA pre-trained model are also important, using the BART weights is much better than random initialization
> >
> > For a fair comparison, [Unifying multimodal transformer for bidirectional image and text generation. Huang et al. (2021)] mentioned in the table has the somewhat similar idea with OFA but only trained on the MSCOCO. It can be seen that our model outperforms the autoregressive model both in terms of the quality of image generation and the quality of text generation.
> >
> > ### Finally, we would like to thank the reviewer for raising concerns. We will add relevant content to the manuscript to improve the readability of the article.
> >
> > [1] Austin, Jacob, et al. "Structured denoising diffusion models in discrete state-spaces." Advances in Neural Information Processing Systems 34 (2021): 17981-17993.
> >
> > [2] Gu, Shuyang, et al. "Vector quantized diffusion model for text-to-image synthesis." Proceedings of the IEEE/CVF Conference on Computer Vision and Pattern Recognition. 2022.
> >
> > [3] Raffel, Colin, et al. "Exploring the limits of transfer learning with a unified text-to-text transformer." J. Mach. Learn. Res. 21.140 (2020)
> >
> > [4] Saharia, Chitwan, et al. "Photorealistic Text-to-Image Diffusion Models with Deep Language Understanding." arXiv preprint arXiv:2205.11487 (2022).

---

> > > ### Author Response · Authors · 2022-11-30
> > > **Your feedback on our response will be highly appreciated.**
> > >
> > > Dear reviewer cr8c,
> > >
> > > We would like to thank you again for your insightful comments and recommendations. In our earlier responses, we have made every effort to answer your queries and have amended the document in light of your comments. We eagerly await your response to our replies, and we welcome any suggestions for improving our effort.
> > >
> > >
> > > Best.

---

> > > > ### Comment · Reviewer_cr8c · 2022-12-11
> > > > **Rebuttal response**
> > > >
> > > > I would like to thank the authors for their detailed response to my review -- I sincerely appreciate their efforts. These responses have answered my questions raised in the review and I am raising my score to 6.

---

### Author Response · Authors · 2022-11-10
**Summary Response**

At the outset, I would want to present two somewhat rigorous concepts to make subsequent explanations easier to understand.
- “Modality Translation”: First consider an Image Translation task, in which a source image is converted to the target image, while retaining sufficient key features of the original image. We want to extend this concept more broadly, to one involving multiple modalities. Current multi-modal generation tasks need at least one given conditional modality (for example, text prompt in T2I tasks), and this modality can be regarded as the source domain. The modality to be generated (such as images in T2I tasks) is similar to the target domain. Such multi-modal approaches transfer the features from the source domain to the target domain, thus the source domain features must be provided for the conditional generation. Here we would like to call such multi-modal generative models as “Modality Translation Generative Models”, which include DALLE series [1,2], Parti [3], Imagen [4], VQ-Diffusion [5], Stable Diffusion [6] etc. Mathematically, they estimate a marginal distribution $p(x_{img}|x_{txt})$.
- “Modality Generation”: The unconditional generative models, like DDPM, VAEs and GANs etc. are able to directly generate content within a single modality, e.g. images of faces or churches. These do not require any prior conditional input. Hence we can call these unconditional models as “Modality Generative Models". Such models construct a distribution of $p(x_{img})$.
- We would therefore like to define our model as a “Multi-Modality Generative Model”. To the best of our knowledge, almost no generative model has been able to create multimodal content concurrently in the absence of any prior conditional info. To fill this gap, we proposed our work, a discrete diffusion based generative model, to solve the “Multi-Modalities Generation” Tasks. Proposed Multi-Modality Generative Model also has potential in terms of:
    1. Our work can generate multi-modal content at the same time without given conditions using single model. And benefitting from such property, we can also achieve the manipulation on different modalities as well as the Modality Translation Tasks with the same model.
    2. Our work can learn a more compact connection within different modalities.
    3. Our work offers an access to the joint distribution of both image and text $p(x_{img},x_{txt})$.
- As mentioned by Reviewer tuqJ, it should be noted that current measurements do not entirely represent which model is superior, and the visual results are on par with existing methods, for example, the FID score is a metric that below some level is indeed indicating a dataset replication.

We have revised the manuscript according to all the reviewers’ comments. The changes are highlighted in purple. We have emphasised the contribution of the article and added some mathematical expressions of our proposed methods. We clarify how our solutions differ from traditional solutions by defining the “modality translation” and “modality generation”. The limitations and shortcomings of our existing methods are also discussed in the appendix, and potential solutions are given.

We now return to the raised concerns.

[1] Ramesh, Aditya, et al. "Zero-shot text-to-image generation." International Conference on Machine Learning. PMLR, 2021.

[2] Ramesh, Aditya, et al. "Hierarchical text-conditional image generation with clip latents." arXiv preprint arXiv:2204.06125 (2022).

[3] Yu, Jiahui, et al. "Scaling autoregressive models for content-rich text-to-image generation." arXiv preprint arXiv:2206.10789 (2022).

[4] Saharia, Chitwan, et al. "Photorealistic Text-to-Image Diffusion Models with Deep Language Understanding." arXiv preprint arXiv:2205.11487 (2022).

[5] Gu, Shuyang, et al. "Vector quantized diffusion model for text-to-image synthesis." Proceedings of the IEEE/CVF Conference on Computer Vision and Pattern Recognition. 2022.

[6] Rombach, Robin, et al. "High-resolution image synthesis with latent diffusion models." Proceedings of the IEEE/CVF Conference on Computer Vision and Pattern Recognition. 2022.

---

### Comment · Area_Chair_qWPZ · 2022-12-09
**Some questions**

Dear authors,

Could you please clarify the procedure used to accomplish results in Figure 5? Specifically, are the autoencoders trained with clean or masked images / texts? Where do you perform masking for the inpainting tasks (in the image directly or in the encoded image/text tokens)? Are there requirements on what the mask area should look like? Does it support any rectangle masking area of the image up to the image resolutions?

Thanks,
AC

---

> ### Author Response · Authors · 2022-12-09
> **Reply to further questions.**
>
> Dear AC,
>
> Thanks for your questions.
>
> **Could you please clarify the procedure used to accomplish results in Figure 5?**
>
> Given an image $x$ and a mask $M$, we set all pixels under mask area $M$ as $0$ and get a masked image $\hat{x}$. Then the masked image $\hat{x}$ is fed into a pre-trained quantised VAE and can be represented in discrete index. Our model will automatically fill up the masked area in $\hat{x}$.
>
> **Specifically, are the autoencoders trained with clean or masked images / texts?**
>
> The quantised autoencoders are trained with clean images. And Byte Pair Encoding (BPE) are obtained within clean texts.
>
> **Where do you perform masking for the inpainting tasks (in the image directly or in the encoded image/text tokens)?**
>
> We directly perform masking in the image pixel domain.
>
> **Are there requirements on what the mask area should look like?**
>
> There’s no requirement for the masked area shape.
>
> **Does it support any rectangle masking area of the image up to the image resolutions?**
>
> Yes, we can use any size of masks.
>
> Best,
> Authors

---

> > ### Comment · Area_Chair_qWPZ · 2022-12-09
> > **Follow-up**
> >
> > Thanks for the clarification. It is still not clear to me how "our model will automatically fill up the masked area in $\hat{x}$" works? What exactly are performed on those VQ-VAE tokens in order to obtain a complete image (with masked area filled in)? Furthermore, since the VQ-VAE is trained on clean image/text, isn't it problematic to take a masked image/text as its input during the test time? Have you notice any artifacts when you modify the mask size and shape?

---

> > > ### Author Response · Authors · 2022-12-10
> > > **Reply to following up questions from AC qWPZ**
> > >
> > > Thanks for your questions.
> > >
> > > **What exactly are performed on those VQ-VAE tokens in order to obtain a complete image (with masked area filled in)?**
> > >
> > > 1. Image $x$ in pixel domain with mask converts to discrete domain $\hat{x}$ with a VQ-VAE.
> > > 2. The token under mask area will be set as $\texttt{[MASK]}$. (as the pixels are all $0$ and the discrete token index for such area will be unique.)
> > > 3. The Discrete Diffusion model will fill up the $\texttt{[MASK]}$ tokens during denosing process.
> > >
> > >
> > > **Since the VQ-VAE is trained on clean image/text, isn't it problematic to take a masked image/text as its input during the test time?**
> > >
> > > The denoising and diffusion process are conducted in discrete domain. Thus the VQ-VAE only need to process the clean image during training. Only for the inpainting or manipulation tasks, the VQ-VAE take the masked images. However, it can successfully encode the image into the discrete domain.
> > >
> > > **Have you notice any artifacts when you modify the mask size and shape?**
> > >
> > > Yes, there will be some artifacts in *reconstructed* images during the change of the shape and size of masks. But there is no artifacts in *generated* images.
> > >
> > > *reconstructed* images mean we directly decode the image with $\texttt{[MASK}]$ token, while *generated* images are obtained after denoising process.
> > >
> > >
> > > Best,
> > > Authors.

---

> > > > ### Comment · Area_Chair_qWPZ · 2022-12-12
> > > > **contradiction?**
> > > >
> > > > Thanks for the clarification. I understood the inpainting process now. But if you set the discrete tokens to [mask] for masked area, this sounds like you have to mask image area that is exactly aligned with discrete tokens, not what you mentioned "There’s no requirement for the masked area shape". For example, say you have 128x128 pixels -> 32x32 tokens, each discrete token corresponding to 4x4 patches, if you only mask part of 4x4 patches in pixels, you have to set its token to [mask] which it is like masking the whole patch, right? Did I miss something?

---

> > > > > ### Author Response · Authors · 2022-12-12
> > > > > **Further explanation**
> > > > >
> > > > > We can explain the inpainting process in more detail with the initial understanding.
> > > > >
> > > > > Regardless of the shape of the mask in the pixel domain, the representation in the discrete domain is rectangle.
> > > > >
> > > > > *  Based on the assumptions, if the 4x4 pixels we mask at pixel domain happen to correspond to a discrete token, then this discrete token can be easily set to $\texttt{[MASK]}$.
> > > > > *  If we still mask the 4x4 pixels but he involves multiple tokens in multiple discrete fields, then the contents of all these tokens need to be **diffused**.
> > > > > *  Although the pixel area corresponding to multiple tokens may be larger than the pixel area of the mask in cases where multiple tokens are involved, we can use $I_{out}=(1-m) \times I_{out} + mask \times I_{in}$ to get the final output and maintain the original visible contents. Here $I_{out}$ is the decoded image in the pixel domain and $m$ is the mask in pixel domain.
> > > > > *  Arbitrarily shaped masks can be considered a variant of the third scenario.
> > > > > *  Thus we assume that there’s no requirement for the masked area shape in pixel domain.
> > > > >
> > > > > Best,
> > > > > Authors

---

> > ### Comment · Reviewer_bEZ6 · 2022-12-09
> > **follow-up question**
> >
> > I have a follow-up question:
> >
> > After getting the VQGAN tokens of masked image $\hat{x}$, shouldn't we set the corresponding tokens to $\texttt{[mask]}$ as well? (otherwise, there will be no "mask" to fill in) If so, the inpainting is actually performed in token space, right? (Although there's no information leakage since image pixels are also masked.)
> >
> > Thanks,
> > Reviewer bEZ6

---

> > > ### Author Response · Authors · 2022-12-10
> > > **Reply to following up questions from Reviewer bEZ6**
> > >
> > > Thanks for your questions, your assumption is correct. Inpainting actually occurs in discrete spaces.
> > >
> > > As we set all the pixels under mask as $0$, the corresponding discrete tokens in such area will be convert to $\texttt{[MASK]}$
> > >
> > > Best,
> > > Authors

---

### Decision · Program_Chairs · 2023-01-20

**Decision:**

Accept: poster

**Justification For Why Not Higher Score:**

There are limitations as mentioned above, so this is a borderline paper.

**Justification For Why Not Lower Score:**

This is a borderline paper, and can be rejected which is fine with AC and reviewers too.

**Metareview: Summary, Strengths And Weaknesses:**

The paper proposes a latent diffusion model for text-image joint modeling. The image and text are encoded into latent space, and they trained a discrete diffusion models that jointly model the discrete latent codes, which are then decoded into images and texts. This allow inpainting of both text and images in both directions. While reviewers expressed this is an interesting and potentially novel application, the reviewers also expressed concerns in following aspects: 1) the experimental results are doing significantly worse than baselines that the paper compares, in both image->text and text->image directions, 2) the proposed technique is only incremental to existing VQ-GAN, 3) the paper writing is not intuitive and unclear on how the method works (such as how inpainting is done). Overall, reviewers and AC agree that this is a borderline paper. At the end of the day, the application is still interesting, pointing towards a promising direction, so I'm leaning a little bit towards acceptance.

**Note From Pc:**

if the above contains the word "oral" or "spotlight" please see: "oral" presentation means -> notable-top-5% and "spotlight" means -> notable-top-25%. As stated in our emails, we are disassociating presentation type from AC recommendations

**Summary Of Ac-Reviewer Meeting:**

During the meeting, following points are mentioned

* The reviewer who gave highest ratings believe that it should be between 6 and 8. They are biased towards 8 as there are too many tough reviews nowadays... They also mentioned there is no other Diffusion models atempting image-text joint modeling at the submission time, so this is a novel problem/task/application.
* Reviewers expressed that the writing of paper is problematic, such as not clear how masking is done.
* Reviewers also mentioned that the evaluation has problems, such as using bigger bigger models for the small dataset.
* Novelty is limited in the method, the proposed mutual attention is similar to cross attention in VQ-GAN.